# Assessment of WRF (v 4.2.1) dynamically downscaled precipitation on subdaily and daily timescales over CONUS

Abhishekh Kumar Srivastava[1], Paul Aaron Ullrich[1], Deeksha Rastogi[2], Pouya Vahmani[3], Andrew Jones[3], and Richard Grotjahn[1]

[1]Department of Land, Air and Water Resources, University of California, Davis, CA, USA.
[2]Computational Science and Engineering Division, Oak Ridge National Laboratory, Oak Ridge, Tennessee, USA.
[3]Climate and Ecosystem Sciences Division, Lawrence Berkeley National Laboratory, CA, USA.

**Correspondence:** Abhishekh Kumar Srivastava (asrivas@ucdavis.edu)

**Abstract.** This study analyzes the quality of simulated historical precipitation across the contiguous United States (CONUS) in a 12-km Weather Research and Forecasting model version 4.2.1 (WRF v 4.2.1)-based dynamical downscaling of the fifth-generation ECMWF atmospheric reanalysis (ERA5). This work addresses the following questions: First, how well are the 3- and 24-hr precipitation characteristics (diurnal and annual cycles, precipitation frequency, annual and seasonal mean and

maximum precipitation, and distribution of seasonal maximum precipitation) represented in the downscaled simulation, compared to ERA5? And second, how does the performance of the simulated WRF precipitation vary across seasons, regions, and timescales? Performance is measured against the NCEP/EMC 4-km Stage IV and PRISM data on 3-hr and 24-hr timescales, respectively. Our analysis suggests that the 12-km WRF exhibits biases typically found in other WRF simulations, including those at convection-permitting scales. In particular, WRF simulates both the timing and magnitude of the summer diurnal pre-

cipitation peak as well as ERA5 over most of the CONUS, except for a delayed diurnal peak over the Great Plains. As compared to ERA5, both the month and the magnitude of the precipitation peak annual cycle are remarkably improved in the downscaled WRF simulation. WRF slightly overestimates 3- and 24-hr precipitation maximum over the CONUS, in contrast to ERA5 which generally underestimates these quantities mainly over eastern half of the CONUS. Notably, WRF better captures the probability density distribution (PDF) of 3- and 24-hr annual and seasonal maximum precipitation. WRF exhibits seasonally-

dependent precipitation biases across the CONUS, while ERA5's biases are relatively consistent year-round over most of the CONUS. These results suggest that dynamical downscaling to a higher resolution improves upon some precipitation metrics, but is susceptible to common regional climate model biases. Consequently, if used as input data for domain-specific models, we suggest moderate bias-correction be applied to the dynamically downscaled product.

 # 1   Introduction

Dynamical downscaling refers to the use of regional climate models forced with initial and lateral boundary conditions derived from either a global climate model or reanalysis to generate high-resolution climate output (Giorgi and Mearns, 1991). These high-resolution simulations add value through better representation of regional weather and climate phenomena, especially over regions of complex and heterogeneous topography (Doblas-Reyes et al., 2021). For example, better representation of local topography, water bodies and land-sea contrast improves local scale processes such as fine scale convection, land-sea breeze, and nonlinear interactions between local, mesoscale and large-scale processes (Caldwell et al., 2009; Di Luca et al., 2015; Ashfaq et al., 2016; Prein et al., 2016; Bozkurt et al., 2019; Rastogi et al., 2022). Both the higher resolution and improved representation of physical processes facilitate the study of future changes in the mean and variability of the weather and climate systems (Barsugli et al., 2013) and distilled user-oriented regional climate information on local and regional scales (Rhoades et al., 2020; Doblas-Reyes et al., 2021; Ranasinghe et al., 2021).

Though increased resolution in downscaled climate models is fundamentally important for their utility at regional scales, it is not sufficient for ensuring reliable and accurate information. The biases in regional climate model output are well documented. These biases can originate from various sources, including the lateral boundary conditions (Christensen et al., 2008; Schoetter et al., 2012; Giorgi, 2019) and parameterization schemes (Iguchi et al., 2017; Kong et al., 2022). The biases may also vary with variable, region and season of interest (Castro et al., 2005; Prein et al., 2015; Diaconescu et al., 2016; Srivastava et al., 2020, 2021, 2022). High resolution and high quality climate data have many uses for both advancing process-understanding and for informing operations, particularly at local to regional scales. The ECMWF atmospheric reanalysis (ERA5; Hersbach et al., 2020) represents a great stride forward in the development of a complete historical meteorological dataset with sufficiently high temporal and spatial resolution to represent many forms of extreme weather and their impacts. However, for investigating water resource availability, mountain snowpack, and land-atmosphere fluxes, particularly in the context of multi-sectoral dynamics, even finer grid spacing is required. Given the broad interest among scientists and stakeholders in developing regional climate data products at 1/8° grid spacing based on high-quality reanalysis, it is important to investigate to what degree (if any) the dynamically downscaled data improves upon the original ERA5 product. Such a study is further valuable for informing other dynamical downscaling efforts, such as the international Coordinated Regional Downscaling Experiment (CORDEX) program (Gutowski Jr. et al., 2016).

In this study we evaluate historical precipitation over the contiguous United States (CONUS) in a 12-km Weather Research and Forecasting model version 4.2.1 (WRF v 4.2.1)-based dynamical downscaling of ERA5 over the period 1980-2020. This WRF-based historical simulation is part of an ensemble data product (Jones et al., 2022) that includes thermodynamic global warming (TGW) simulations under projected climate forcings (Jones et al., under review). In this paper, we specifically ask: (1) How well are the 3- and 24-hr precipitation characteristics (diurnal and annual cycles, precipitation frequency, annual and seasonal mean, and maximum precipitation, and distribution of seasonal maximum precipitation) represented in the downscaled WRF simulation, in comparison to ERA5? (2) How does the performance of the simulated WRF precipitation vary across seasons, regions, and timescales? The performance of 3-hr ERA5 and WRF precipitation simulations are measured against the

NCEP/EMC 4KM Gridded Stage IV Data (Stage IV). The performance of 24-hr ERA5 and WRF precipitation simulations are measured against the Oregon State University Parameter-Elevation Regressions on Independent Slopes Model (PRISM) dataset.

The specific questions above are motivated by several important considerations. Most previous studies have focused on the accuracy of the simulated precipitation on daily or longer timescales (e.g., Bukovsky and Karoly, 2009; Caldwell et al., 2009; Rhoades et al., 2020; Srivastava et al., 2021, 2022; Gensini et al., 2022), likely because of availability of data on daily timescales. However, many of the most high-impact precipitation-related physical processes (such as short-duration convective storms leading to extreme precipitation events or precipitation intermittency) occur at hourly timescales (Westra et al., 2013; Trenberth et al., 2017), and conclusions drawn from analyzing longer time-scale precipitation do not automatically translate to shorter timescales (Barbero et al., 2019). Further, regional climate models are known to be sensitive to both the resolved (e.g., horizontal resolution and simulation domain) and unresolved parameters (e.g., convection parameterization schemes), and so particular regional climate model configurations must be examined before they can be used for regional application (Giorgi and Mearns, 1999; Liang et al., 2004). The 12km WRF simulation examined in this study uses a convective parameterization, which is considered to be a major source of model biases on both subdaily and daily timescales (Dirmeyer et al., 2012; Hanel and Buishand, 2010; Knist et al., 2020). Moreover, a seasonal analysis of precipitation is important as, generally, both the observation-based datasets (e.g., reanalyses) and models (e.g., WRF) better simulate precipitation in winter than in summer, mainly because winter precipitation is mostly dominated by predictable large scale stratiform systems (Ebert et al., 2007) and summer precipitation is mainly influenced by unpredictable small-scale convective cells (Prein et al., 2015; Beck et al., 2019).

The rest of the paper is organized as follows: Section 2 describes the data and methodology used. Results are presented and discussed in section 3, and also tabulated in Tables 2 and 3, then summarized in section 4. Figures for percent biases are included in the Supplementary Material.

## 2 Data and Method

### 2.1 Data

#### 2.1.1 WRF downscaling of ERA5

The Weather Research and Forecasting model version 4.2.1 is a state-of-the-art, fully compressible, non-hydrostatic, mesoscale numerical weather prediction system designed for both atmospheric research and operational forecasting applications (Ska-marock et al., 2008). For this study, the WRF simulation is carried out at 12-km horizontal grid spacing and covers the 1980-2020 period (Fig. 1). The physical parameterizations chosen are: Thomson microphysics (Thompson and Eidhammer, 2014), the Tiedke cumulus parameterization (Tiedtke, 1989; Zhang et al., 2011), the Mellor-Yamada-Janjic boundary layer scheme (Janjić, 1994), and the Eta similarity surface layer (Janjić, 1994). Noah is employed for modeling the land surface (Tewari et al., 2004). WRF is further coupled with an urban canopy model (UCM), which resolves urban surfaces, and its land use/land cover is based on National Land Cover Data (NLCD, Dewitz, 2021). Studies suggest that urbanization can enhance or suppress

precipitation over different regions, situations, and urbanization phases. Some examples are: Wang et al. (2015) show that urban warming during the early urbanization phase promotes increased sensible heat flux, enhanced convergence, and vertical motion, leading to urban modification of rainfall. Li et al. (2022) find that urbanization suppresses summer precipitation from mesoscale convective systems, isolated deep convection, and non-convective systems in the Mid-Atlantic region east of the Rocky Mountains. Georgescu et al. (2021) report that physical growth of the built environment can either enhance or suppress extreme precipitation across CONUS metropolitan regions.

The initial and boundary conditions are obtained from the ERA5 dataset (Hersbach et al., 2020). ERA5 is a fifth-generation ECMWF reanalysis product that assimilates a suite of observations (e.g., aircraft, in situ, and satellite) into the Integrated Forecasting System (IFS) to produce hourly meteorological variables on a regular 0.25 degrees lat-lon grid with 137 vertical levels.

### 2.1.2 Reference datasets

To evaluate the performance of 3-hr WRF precipitation, NCEP/EMC 4KM Gridded Data Stage IV Data (Stage IV) is used as reference (Lin and Mitchell, 2005). Stage IV is available at hourly temporal resolution and at 4 km horizontal grid spacing. Stage IV is generated at NCEP from the regional hourly and 6-hourly multi-sensor (radar + gauges) precipitation analyses produced by the 12 River Forecast Centers (RFCs) over the Continental United States. Beck et al. (2019) report that, to minimize systematic biases in Stage IV data, the dataset is rescaled to match its long-term mean with that of the PRISM dataset (details given below) over the evaluation period (2008–2017).

The performance of 24-hr WRF precipitation is evaluated against the Oregon State University Parameter-Elevation Regressions on Independent Slopes Model (PRISM) dataset at 4km grid spacing (Daly et al., 2008). The daily PRISM data uses in situ data with a digital elevation model to account for the complex meteorological response from orography, rain shadows, temperature inversions, slope aspect, coastal proximity, and other local features.

For comparison, ERA5, Stage IV and PRISM precipitation datasets are interpolated to the 12-km WRF grid using first-order conservative remapping (Jones, 1999).

### 2.2 Method

In this study, we estimate precipitation metrics that characterize the frequency, total amount, intensity, and timing of the mean and extreme precipitation. The metrics are summarized in Table 1. We calculate the mean precipitation amount for 3- and 24-hr durations using all precipitation values, including zeros. We use 0.25mm and 1mm thresholds for estimating the frequency and mean precipitation during wet 3-hr and 24-hr periods, respectively. We use these thresholds to minimize the effect of excessive drizzle being present in regional climate models and reanalyses (e.g., Frei et al., 2006; Rajczak et al., 2013), and also to account for observational constraints (Schär et al., 2016). The differences between the mean precipitation amount and the mean wet-3-hr/ wet-24-hr precipitation highlight the biases that result from excessive drizzle in the dataset. The precipitation thresholds in the study are consistent with those in previous studies (e.g., Rajczak et al., 2013; Rajczak and Schär, 2017; Xiao et al., 2018; Kooperman et al., 2022).

### 2.2.1 Diurnal and annual cycle of precipitation

The diurnal cycle of precipitation is estimated by fitting the first two harmonics to the monthly mean 3-hr precipitation. Similarly, the annual cycle of precipitation is estimated by fitting the first two harmonics to the monthly mean 24-hr precipitation. The timing of the diurnal peak of the 3-hr precipitation is expressed in terms of local solar time (LST). 12 noon LST is the time when the Sun is highest in the sky at a location. LST hours are obtained from UTC hours as follows (Watters et al., 2021):

$$t_{LST} = t_{UTC} + \frac{\lambda^\circ}{15^\circ h^{-1}}, \tag{1}$$

where, $t_{UTC}$ and $t_{LST}$ are the coordinated universal time and local solar time, respectively. $\lambda$ is the longitude, in degrees.

In this work, the subdaily precipitation is examined for the 2003-2019 period and the daily precipitation is analyzed for the 2001-2020 period. These periods are chosen for two considerations. First, the hourly Stage IV data are available only after 2002. Second, any variability arising from the trend may be assumed to be insignificant in the 20-year record. The results are summarized for the seven National Climate Assessment (NCA) regions over the CONUS (https://www.globalchange.gov/content/nca5-regions). The seven NCA regions are: NW (northwest), SW (southwest), NGP (northern Great Plains), SGP (southern Great Plains), MW (Midwest), SE (southeast), and NE (northeast).

## 3 Results

### 3.1 Diurnal cycle of precipitation

Fig. 2 shows the peak time of the JJA diurnal precipitation peak (TDPP) in ERA5, WRF and Stage IV datasets (in hours at LST). We chose to analyze the JJA diurnal cycle because the diurnal variations are stronger in summer than in winter (Dai et al., 1999). Presumably, this is because winter variations of precipitation are dominated by frontal cyclones, and a frontal passage can occur at any time of day, thereby masking any diurnal cycle present. During summer, the frontal cyclone passages are much less frequent, allowing the diurnal cycle to be more visible (e.g., Kunkel et al., 2012). The observed (Stage IV) spatial pattern of TDPP shows that, mostly, precipitation peaks in the afternoon over most of the CONUS, except for regions to the east of the Rocky Mountains (the Great Plains and MW regions). The eastward propagating shift in nighttime diurnal peak east of the Rockies is consistent with mesoscale convective systems (MCSs) originating over the Rockies and moving eastward (Dai et al., 1999; Tan et al., 2019; Scaff et al., 2020; Watters et al., 2021). ERA5 generally reproduces the spatial pattern of the observed diurnal cycle, but the peak occurs earlier along the northern boundaries of the Northern Great Plains (NGP) and west of the Great Lakes in the Midwest (MW). The largest biases in ERA5 are found between 100°-85°W, also noted in Watters et al. (2021) who compared biases in ERA5 against the Multi-Radar Multi-Sensor (MRMS) gauge-adjusted ground-based radar network product. Similar to ERA5, WRF simulates the observed timing of the diurnal precipitation peak everywhere except over the regions east of the Rockies. Over the regions falling east of 100°W, the observed late night to early morning peak in the diurnal cycle is delayed in the WRF simulation. Similar behavior was also noted in the convection-permitting WRF simulation

of Scaff et al. (2020). The slow propagation eastward of convective systems is driven by cloud-scale phenomena that are not necessarily well captured by the models used to generate datasets.

The observed magnitude of the JJA precipitation diurnal cycle (using MDPP; precipitation magnitude during the peak of the diurnal cycle) is larger in the eastern CONUS compared to the western CONUS (Fig. 3). The largest MDPP magnitudes are observed along the Gulf coast and in Florida. ERA5 simulates the observed spatial pattern of the diurnal precipitation magnitude very well. Watters et al. (2021) found that ERA5 generally overestimates the magnitude over much of the CONUS in comparison to the Integrated Multi-satellitE Retrievals for GPM (IMERG) dataset, possibly due to reliance on the convection parameterization. The differing performance of ERA5 against the two different observational datasets (as noted in Watters et al. (2021) and our study) also points to uncertainties arising due to differences in reference datasets. WRF does capture the spatial pattern of the observed diurnal precipitation peak magnitude over most of the CONUS; except over the Southeast where it overestimates the magnitude of the precipitation peak, and over the central Great Plains region, where it underestimates the magnitude more than ERA5. The dry biases over the midwest and parts of the central CONUS in WRF diurnal precipitation magnitude are consistent with those of Scaff et al. (2020), suggesting that current climate models, including WRF, underestimate MCS frequencies in summertime weak synoptic-scale forced conditions (Prein et al., 2020). The wet MDPP bias in WRF over the SE is also observed in previous WRF-based studies (e.g., Wang and Kotamarthi, 2014; Scaff et al., 2020). Sun and Bi (2019) showed that the WRF simulation with the Tiedke cumulus parameterization scheme exhibits an earlier and stronger diurnal cycle than the observed over land regions between $25°S$ and $25°N$ in boreal summer. As the convective scheme is the most crucial model component in capturing the diurnal cycle of precipitation (Shin et al., 2007); and precipitation from cumulus parameterization schemes dominates over the SE CONUS (Iguchi et al., 2017), we suspect that cumulus parameterization in the current WRF simulation may be responsible for the wet bias over the SE region.

## 3.2 Annual cycle of precipitation

Fig. 4 shows the peak time (calendar month) of the monthly averaged precipitation (TMPP; the annual cycle of the monthly averaged precipitation). Using PRISM as a reference, maximum monthly precipitation occurs during winter season over the western CONUS and parts of Arkansas, Mississippi, Louisiana, and the NE CONUS. The majority of the Great Plains is dominated by the late spring and early summer precipitation, whereas the Southeast region gets most of the rainfall in the summer season. This high-resolution spatial map of the annual cycle of monthly precipitation is consistent with previous studies (e.g., Bukovsky and Karoly, 2007). Stage IV also exhibits a similar annual precipitation cycle as PRISM; however, differences from PRISM emerge over multiple regions across the US such as SW (Utah) , NE (Maine), and MW (northern and eastern boundaries of Lake Michigan). A few sources of biases in Stage IV may affect its results shown in Fig. 4 and subsequent figures. For e.g., a discontinuity in the mosaic-making process exists over a few regions such as over oceans, and areas that cover the Great Lakes region. A few western RFCs, including Colorado Basin RFC (CBRFC) do not use radar estimates due to poor coverage over mountainous regions. Moreover, the inherent biases in radar rainfall estimation due to factors such as lack of radar coverage, brightband contamination, and biases existing in the algorithms, are not completely avoidable (Nelson et al., 2016; Prat and Nelson, 2015). Both ERA5 and WRF are able to simulate the spatial pattern of peak time of the annual

cycle. However, WRF outperforms ERA5 in simulating the spatial structure of the annual cycle, as it greatly improves ERA5 biases over the NE, and parts of the SE and Great Plains regions.

The spatial pattern of the magnitude of the monthly averaged precipitation peak (MMPP) is shown in Fig. 5. The maximum monthly average precipitation occurs along the western coast, Sierra Nevada mountains and in the Southeastern region. Stage IV does capture the spatial pattern of the referenced precipitation magnitude; it exhibits underestimated precipitation (dry bias) of 20% or more almost everywhere across the CONUS. The largest percent biases exist over the SE and SW regions (Fig. 5 and Supplementary Fig. S2). ERA5 underestimates the precipitation magnitude over the NE and SE regions, and overestimates it over the Southern Great Plains. On the other hand, WRF captures the spatial pattern of the magnitude very well across CONUS, and exhibits much lower biases across the CONUS than ERA5.

In summary, both the timing and magnitude of the monthly averaged precipitation peak are improved in the downscaled WRF simulations compared to ERA5.

### 3.3 Evaluation of 3-hr precipitation

Fig. 6 shows the precipitation frequency of 3-hr precipitation (PF3h). The precipitation frequency is computed as the counts of 3-hr precipitation events with magnitude greater than 0.25 mm expressed as a percentage of the total number of 3-hr time steps. Compared with Stage IV, ERA5 overestimates the precipitation frequency by 3–10% in all seasons over most of the CONUS except over the NW and SW. It does underestimate the frequency over the hilly areas of the NW regions in JJA. WRF also exhibits more frequent precipitation mostly over NGP and MW regions in DJF and MAM. In contrast, WRF consistently underestimates precipitation frequency along the west coast. Also, notably, WRF overestimates the frequency over the SE in MAM and JJA and underestimates it in DJF. The spatial pattern of biases in the annual 3-hr precipitation frequency in WRF is consistent with Kong et al. (2022), who found that precipitation frequency in WRF is more sensitive to the convective and radiation schemes than the precipitation amount.

Pmean3h (the mean of all 3-hr precipitation values including zeros) is shown in Fig. 7. In Stage IV data, the 3-hr mean precipitation is maximum over the coastal and mountainous regions of the western US (Washington, Oregon, and Sierra Mountains of California). The eastern half of the CONUS experiences more 3-hr average precipitation than the western half (except in the coastal and mountainous regions). The maximum values of Stage IV 3-hr precipitation observed along the northwestern US states are missing in the satellite-derived and bias-corrected gridded Climate Prediction Center Morphing technique (CMORPH) dataset, probably due to the insufficient representation of orography at $0.25° \times 0.25°$ grid spacing (Kong et al., 2022). ERA5 generally overestimates the 3-hr mean precipitation over much of the CONUS throughout the year. On the other hand, while its performance is an improvement in many regions, WRF overestimates the precipitation over most of the CONUS (except SGP) annually or in winter and spring seasons. When compared across seasons, WRF underestimates the summer precipitation but overestimates the winter precipitation over the SGP region. Moreover, WRF simulates a much larger wet bias over the SE in summer than in any other season. The spatial pattern of the WRF simulated precipitation frequency is similar to the mean precipitation amount, suggesting that the subdaily precipitation frequency affects the corresponding subdaily mean precipitation in WRF. The spatial pattern of annual dry bias in the SGP and wet bias in the SE region is also found in the other

WRF simulation employing the Tiedke cumulus parameterization scheme along with the Rapid Radiative Transfer Model for global models (RRTMG) radiation scheme (Kong et al., 2022).

Fig. 8 shows the 3-hr mean for precipitation greater than 0.25mm/3hr (S3hII). As shown for Stage IV, mean S3hII values are generally higher than Pmean3h across the CONUS. The highest S3hII values are observed over the SE and SGP regions, suggesting that 3-hr precipitation in these regions is dominated by drizzling precipitation ($< 0.25$ mm). Notably, except for parts of NGP, NW, and SW regions in DJF, ERA5 underestimates the mean S3hII over most of the CONUS in all seasons. This ERA5 bias, together with those shown in Figs. 6 and 7 suggest that ERA5 suffers from drizzling effect, causing it to precipitate more frequently but in lesser amounts when it rains. In contrast to ERA5, WRF simulates more S3hII values across the CONUS in DJF and less in JJA. Notably, the absolute S3hII biases in WRF are generally lower than those in ERA5 in most of the seasons and regions.

The spatial pattern of the 3-hr annual maximum precipitation (Rx3h) is shown in Fig. 9. Rx3h in Stage IV exhibits higher values in the eastern half of the CONUS than in the western half. The spatial pattern and the magnitude in Stage IV is similar to that obtained from the Next-Generation Radar (NEXRAD) dataset in Wehner et al. (2021). ERA5 generally underestimates (mostly within $\pm 5$mm) the maximum precipitation in all seasons and everywhere. On the other hand, WRF overestimates the 3-hr annual maximum precipitation over the eastern half of the CONUS, but shows clear seasonal variation in its biases over the western CONUS regions. For example, WRF slightly overestimates the precipitation maxima over parts of the NW, SW, and the GP regions in DJF, but underestimates the maxima over those regions in JJA. A detailed investigation of biases in WRF is out of the scope of this paper, but we suspect that WRF biases in the Great Plains may be attributed to underestimated MCS frequencies (Prein et al., 2020), imperfect cumulus parameterization scheme and biases in the representation of intensity, location, and diurnal cycle of the low-level jet in 12-km WRF simulation (Lee et al., 2017).

The above analysis of average 3-hr annual maximum precipitation provides little information on whether the datasets reasonably simulate the distribution of the 3-hr annual maximum precipitation. Fig. 10 shows the probability density function (PDF) of the 3-hr annual maximum precipitation. In each panel, the y-axis uses a log-scale to clearly show higher, less frequent precipitation values. It is apparent from the figures that ERA5 consistently underestimates extreme precipitation values over all NCA regions and across all seasons. WRF generally improves on the biases in ERA5 by producing higher extreme precipitation values and thereby bringing the PDF of extreme precipitation values close to the observed PDF.

## 3.4 Evaluation of 24-hr precipitation

For 24-hr precipitation analysis, we use PRISM as reference data. We also evaluate 24-hr precipitation in Stage IV against PRISM to quantify observational uncertainty.

Fig. 11 shows the 24-hr precipitation frequency (PF24h). The precipitation frequency is computed from the days when 24-hr precipitation is more than 1 mm/day. Stage IV consistently underestimates (in comparison to PRISM) the precipitation frequency over most of the CONUS. The largest biases in Stage IV precipitation frequency are observed over the NGP in winter and SW throughout the year (Supplementary Fig. S7). The underrepresented precipitation frequency in Stage IV may be related to its difficulty in detecting light and frozen precipitation across the CONUS and, most notably, in the western US, because

the precipitation processing system in Stage IV does not distinguish between liquid and frozen hydrometeor types (Smalley et al., 2014). ERA5 consistently overestimates PF24h by more than 5% in all seasons over most of the CONUS except NW and SW regions. It also underestimates the precipitation frequency over the SW region in summer and fall. In contrast, WRF underestimates the frequency in the NW and SW regions, and shows frequency biases in other regions that are seasonally dependent. For example, over the SE, WRF underestimates the frequency in DJF but overestimates it in JJA. Similarly, WRF overestimates the frequency over NGP and MW in DJF, but it underestimates the frequency over those regions in JJA. It is also notable that WRF underestimates the frequency over most of the CONUS in JJA (except SE) and SON. When compared with the biases in 3-hr precipitation frequency (Fig. 6), the spatial pattern of the biases in ERA5 is similar for both 3-hr and 24-hr precipitation. However, the 24-hr precipitation frequency biases in WRF are larger than those for 3-hr precipitation. This suggests that while ERA5 tends to exhibit more drizzle (i.e., low intensity precipitation), WRF generally concentrates precipitation into fewer days of the year than we see in observations.

Biases in 24-hr precipitation mean (Pmean24h) are shown in Fig. 12. Stage IV shows dry bias as compared to PRISM over most of the CONUS in all seasons, except that it shows wet biases over sporadic locations in NW and SW regions. The corresponding percent bias in Pmean24h (Supplementary Fig. S8) indicates large Stage IV relative dry biases in the western CONUS (NGP, NW and SW) in DJF, possibly related to its inability to detect freezing and light precipitation events, as discussed in the previous subsection. ERA5 consistently exhibits a dry bias in 24-hr mean precipitation over the Southeast throughout the year. When compared with the frequency biases in Fig. 11, it appears that although ERA5 precipitates more frequently than PRISM, it precipitates less during wet days than PRISM. ERA5 generally exhibits wet biases over other regions. Over the NE, ERA5 shows dry biases over regions close to the coasts and wet biases over the inland areas – a pattern that may be associated with the insufficient ability of ERA5 parameterizations to produce sea breeze-induced precipitation (Crossett et al., 2020). WRF generally shows dry biases over the Great Plains, exhibiting typical model biases existing in the state-of-the-art climate models (Srivastava et al., 2020). The spatial patterns of 24-hr frequency biases in WRF (Fig. 11) are similar to the 24-hr mean precipitation. When compared with ERA5, WRF shows stronger dry biases over the Great Plains regions, particularly in JJA. It is interesting to note that the spatial pattern of the seasonal 24-hr mean precipitation biases in WRF is quite similar to those simulated by another recent bias-corrected convection-permitting WRF simulation over the CONUS (Gensini et al., 2022) – for example, JJA dry biases in both the studies are spread over most the CONUS. Similarly dry biases over the SE region are quite similar. What is more striking is that the magnitude of the 24-hr mean biases in our study are largely comparable to those in Gensini et al. (2022). The summer dry biases in the Great Plains have been reported in previous analyses of WRF simulations employing convection-permitting or convection-parameterizing configurations (Sun et al., 2016), and in other regional climate models, including WRF (Mearns et al., 2012; Gao et al., 2017). The summer dry biases in the Great Plains may be associated with the unrealistically strong coupling of convection with the surface heating over the Rocky Mountains, and insufficiently resolved and slow propagating mesoscale systems (Mearns et al., 2012; Tripathi and Dominguez, 2013; Hu et al., 2018).

The 24-hr mean wet-day precipitation (SDII) is shown in Fig. 13. As for the biases in Pmean24h (Fig. 12), Stage IV underestimates SDII almost everywhere, but more prominently over the eastern half of the CONUS in all seasons. ERA5

underestimates SDII over the eastern half of the CONUS (parts of NGP, MW, SGP, and NE) across the year. The dry SDII biases, together with the overestimated frequency and mean precipitation in winter and spring over NGP, MW, and SGP, suggest that ERA5 has too-little-and-too-frequent precipitation bias. WRF exhibits wet SDII biases over most of the CONUS in DJF, except in a few places over the SGP and SE. On the other hand, it shows strong dry biases over the SGP and SE during spring
and over the SGP, MW and SE during summer.

Fig. 14 shows biases in 24-hr annual maximum precipitation (Rx1day). As for the other metrics, Stage IV underestimates Rx1day over the eastern half of the CONUS. The dry bias is most pronounced ($\sim 20\%$) over the Great Plains and MW during summer and over the NGP and northeastern parts of SW ($> 50\%$) during winter (Supplementary Material Fig. S10). On the other hand, Rx1day values in Stage IV are very well represented over NW and SW in all seasons except winter. ERA5 shows
strong and significant dry biases over the eastern CONUS throughout the year. The ERA5 wet biases over the western CONUS are smaller than over the eastern half. These pattern are roughly similar to the 3-hr precipitation biases (Fig. 9). WRF generally shows seasonally-dependent biases across CONUS. For example, it shows wet biases during winter and spring but a mix of wet and dry biases (SGP and MW) during summer and fall. When compared with ERA5, it is evident that though WRF reverses the sign of dry bias over most of the eastern CONUS (except parts of the Great Plains), WRF exhibits smaller magnitude of
biases across the CONUS than ERA5.

Finally, the PDF of 24-hr annual maximum precipitation (PDF24h) is shown in Fig. 15. Stage IV represents well the PDF24h over NW and SW. However, it does show problems in capturing the PDF24h over the NGP throughout the year. It is apparent that ERA5 severely underestimates the annual maximum precipitation across the CONUS and throughout the year. WRF does a much better job of simulating the observed distribution as it reduces the biases in ERA5 frequency distribution of 24-hr
annual maximum precipitation for most of the regions and seasons.

For the sake of convenience, the results discussed in this section are also tabulated in Tables 2 and 3.

## 4   Summary and discussion

This paper evaluates the performance of the 12-km Weather Research and Forecasting (WRF) based dynamical downscaling of the fifth generation ECMWF atmospheric reanalysis (ERA5) in simulating the subdaily and daily precipitation characteristics.
In particular, we evaluate diurnal and annual cycles, frequency and mean precipitation, annual maximum precipitation and its distribution. We addressed two questions specifically: (1) How well are the 3- and 24-hr precipitation characteristics represented in the downscaled WRF simulation in comparison to those in ERA5? (2) How does the performance of the simulated WRF precipitation vary across seasons, regions, and timescales? We measure the ERA5 and WRF precipitation simulation against the NCEP/EMC 4KM Stage IV and PRSIM data on 3-hr and 24-hr timescales, respectively.
Our analysis suggests that WRF performs similarly to ERA5 in capturing the timing and magnitude of the JJA 3-hr diurnal precipitation peak over most of the CONUS, except the Great Plains regions. Over the Great Plains, WRF exhibits a diurnal cycle delayed by a few hours, suggesting that the mesoscale convective systems, that originate in the Rockies and travel eastward, are slower in the WRF simulation – a typical model problem found in many previous studies. WRF simulates the

timing (month) and magnitude of the monthly mean 24-hr precipitation annual cycle much better than ERA5. Notably, WRF improves the timing of the annual cycle over the NE, SE and areas surrounding the Gulf of Mexico.

One noticeable difference between ERA5 and WRF is that ERA5 generally displays similar signs of biases (positive or negative) in most of the precipitation characteristics examined throughout the year and across most of the CONUS. However, WRF exhibits seasonally-dependent biases in the precipitation characteristics across the CONUS. For instance, ERA5 overestimates both the frequency and mean of the 3-hr precipitation over most of the CONUS, except over parts of the western CONUS. On the other hand, WRF underestimates the frequency and mean of the 3-hr precipitation over the SE in winter but overestimates these quantities in summer over that region. Similarly, WRF underestimates the mean 3-hr precipitation over the central Great Plains region in summer but not in winter. Also, ERA5 generally underpredicts the 3-hr annual and seasonal maximum precipitation throughout the year over the CONUS, but WRF overestimates it over the eastern CONUS in all seasons. What is interesting is that ERA5 performs poorly in simulating the observed probability distribution of the 3-hr precipitation and thus severely underestimates the observed 3-hr extreme precipitation, but WRF performs quite well in capturing the observed PDF, thereby reducing the biases in ERA5.

Similar to what was found for the 3-hr precipitation, ERA5 does show similar biases in the 24-hr precipitation, but WRF displays regionally- and seasonally dependent biases. WRF overestimates the 24-hr precipitation frequency over most of the CONUS (except, NW and SW). The 12-km WRF generally exhibits seasonally dependent biases also found in the convection-permitting WRF simulation (Gensini et al., 2022). In this analysis, WRF underestimates the frequency throughout the CONUS in SON, but overestimates the frequency over the eastern half of the CONUS in MAM. The underestimated frequency in WRF is more severe in JJA. Similarly, ERA5 underestimates the 24-hr annual maximum precipitation over the eastern half of the CONUS, most notably in the Great Plains and SE regions; whereas, these biases are generally reduced in magnitude in the WRF simulation, but they also occur with a change in the sign. Notably, ERA5 underestimates the 24-hr annual maximum precipitation over the SE, while WRF overestimates it (though by a smaller overall magnitude). As observed for 3-hr precipitation, WRF shows remarkable improvements in the simulated probability distribution of the 24-hr annual maximum precipitation; throughout the CONUS, ERA5 does have problems in capturing the extreme precipitation magnitudes, suggesting that its representation of the strongest precipitation extremes is overly conservative. These results are also summarized in Tables 2 and 3.

This work adds to the literature addressing the value of dynamical downscaling to higher resolution. Our results echo similar past studies, which generally show a mixture of improvement and deterioration in the quality of simulated fields. Although we find that dynamical downscaling with WRF simulates observed precipitation characteristics reasonably well on both the daily and subdaily timescales, improvements do not emerge everywhere. Particularly, WRF exhibits several common biases found in many other models, which are likely suppressed in ERA5 through data assimilation. As hypothesized in this study, WRF does show seasonally- and regionally- dependent biases in precipitation, while ERA5's biases are less seasonal. Nonetheless, WRF greatly improves upon the PDFs of annual maximum precipitation at both 3-hr and 24-hr timescales, and improves on the month and magnitude of the seasonal precipitation cycle. This suggests the WRF product is generally more useful when it comes to its representation of precipitation extremes – which seems to be a consequence of the fact WRF tends to produce generally

flashier precipitation. These results suggest care should be taken in using the WRF simulations for further applications such as
future regional climate projections or regional hydrologic modeling.

A related question is how much bias is acceptable in a climate model. The acceptable level of biases really depends on the application of the climate data. Although the data could be used directly in analysis, we expect a large portion of users will use the data to force other models. In that case, tolerance for biases depends on the type, scope, and scale of the downstream modeling frameworks. Nonetheless, the question is hard to answer quantitatively given that a large uncertainty exists even
among observational datasets (e.g., Srivastava et al., 2020, 2022). Still, one can qualitatively assess the model's performance by comparing it with other models or observational datasets. We assessed the observational uncertainty in 24-hr precipitation representation by comparing precipitation characteristics between PRISM and Stage IV in 24-hr precipitation analysis. We found that biases in WRF are generally smaller in magnitude than in Stage IV. For example, annual 24-hr precipitation frequency (PF24h) is better simulated in WRF than in Stage IV, and biases in the magnitude of monthly average precipitation
peak (MMPP) are much smaller in WRF than in Stage IV. Similarly, WRF shows comparable (e.g., DJF PDF24h in NW and SW) or even better (e.g., NGP in all seasons) simulation of Rx1day PDF (PDF24h) than Stage IV. These analyses suggest that WRF reasonably simulates the observed precipitation characteristics across the CONUS.

While the 12km grid spacing of these simulations is a clear refinement on the native resolution of ERA5, ultimately, it would be far more desirable to run the downscaled simulation in the convection-resolving regime (i.e., 3km or finer). We expect the
match between the precipitation frequency distribution in the tail will improve monotonically with resolution. Until convection-resolving scales are reached, important processes such as horizontal propagation of mesoscale convective systems will not be properly represented. Consequently, when it becomes possible to reach these spatial scales at climatological time scales with available computing power, we would advocate for the metrics explored in this study to be revisited.

*Code and data availability.* The WRF source code is available on GitHub: https://github.com/wrf-model. ERA5 is publicly accessible
from https://www.ecmwf.int/en/forecasts/datasets/reanalysis-datasets/era5. PRISM precipitation data can be downloaded from https://prism. oregonstate.edu/, and Stage IV data is available on https://data.eol.ucar.edu/dataset/21.093. WRF data is accessible at https://data.msdlive.org/records/ksw6 2xv06. The 40-year historical WRF dataset can be downloaded from https://data.msdlive.org/records/ksw6r-2xv06.

*Author contributions.* Abhishekh Kumar Srivastava: Conceptualization; Methodology; Software; Validation; Formal analysis; Investigation; Writing - original draft, review & editing. Paul A. Ullrich: Conceptualization; Validation; Resources; Writing - original draft, review &
editing; Supervision; Project administration; Funding acquisition; Deeksha Rastogi, Pouya Vahamani and Andrew Jones: Model Simulation; Writing - review & editing. Richard Grotjahn: Writing - original draft, review & editing; Supervision.

*Competing interests.* The authors declare that they have no known competing financial interests.

*Acknowledgements.* This work is supported by the Department of Energy Office of Science award number DE-SC0016605, "A Framework for Improving Analysis and Modeling of Earth System and Intersectoral Dynamics at Regional Scales." The climate forcing for this paper was developed collaboratively between the IM3 and HyperFACETS projects, both of which are supported by the U.S. Department of Energy, Office of Science, as part of research in MultiSector Dynamics, and Regional and Global Model Analysis, Earth and Environmental System Modeling Program. A portion of this research used the computing resources of the National Energy Research Scientific Computing Center (NERSC), a U.S. Department of Energy Office of Science User Facility located at Lawrence Berkeley National Laboratory, operated under Contract No. DE-AC02-05CH11231. DR is an employee of UT-Battelle, LLC, under contract DEAC05-00OR22725 with the US Department of Energy (DOE). Accordingly, the publisher, by accepting the article for publication, acknowledges that the US government retains a nonexclusive, paid-up, irrevocable, worldwide license to publish or reproduce the published form of this manuscript, or allow others to do so, for US government purposes. DOE will provide public access to these results of federally sponsored research in accordance with the DOE Public Access Plan (https://www.energy.gov/downloads/doe-public-access-plan).

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

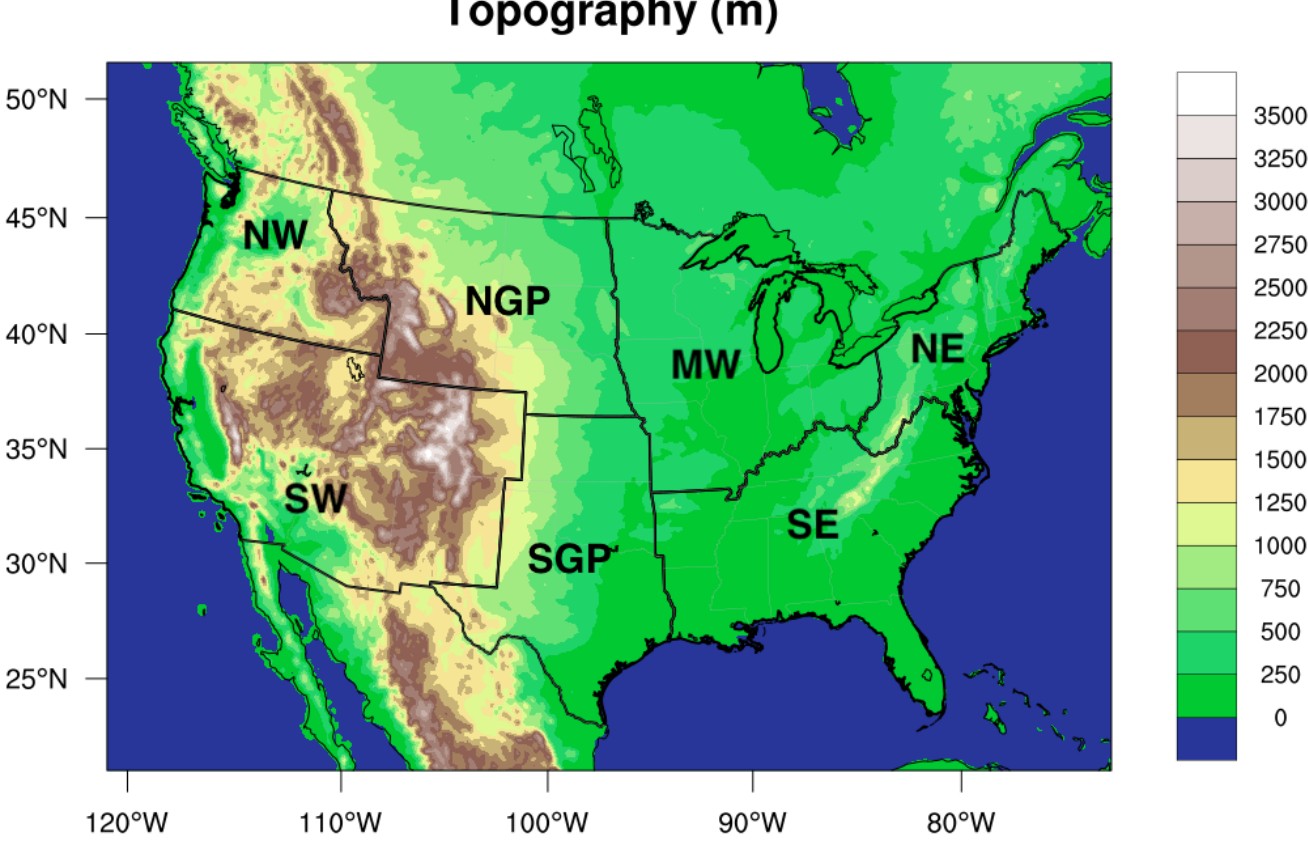

**Figure 1.** The WRF domain employed in this study. Colors denote topography in meters. Bounded regions show the 7 National Climate Assessment (NCA) regions: Northwest (NW), Southwest (SW), Northern Great Plains (NGP), Southern Great Plains (SGP), Midewest (MW), Southeast (SE), Northeast (NE).

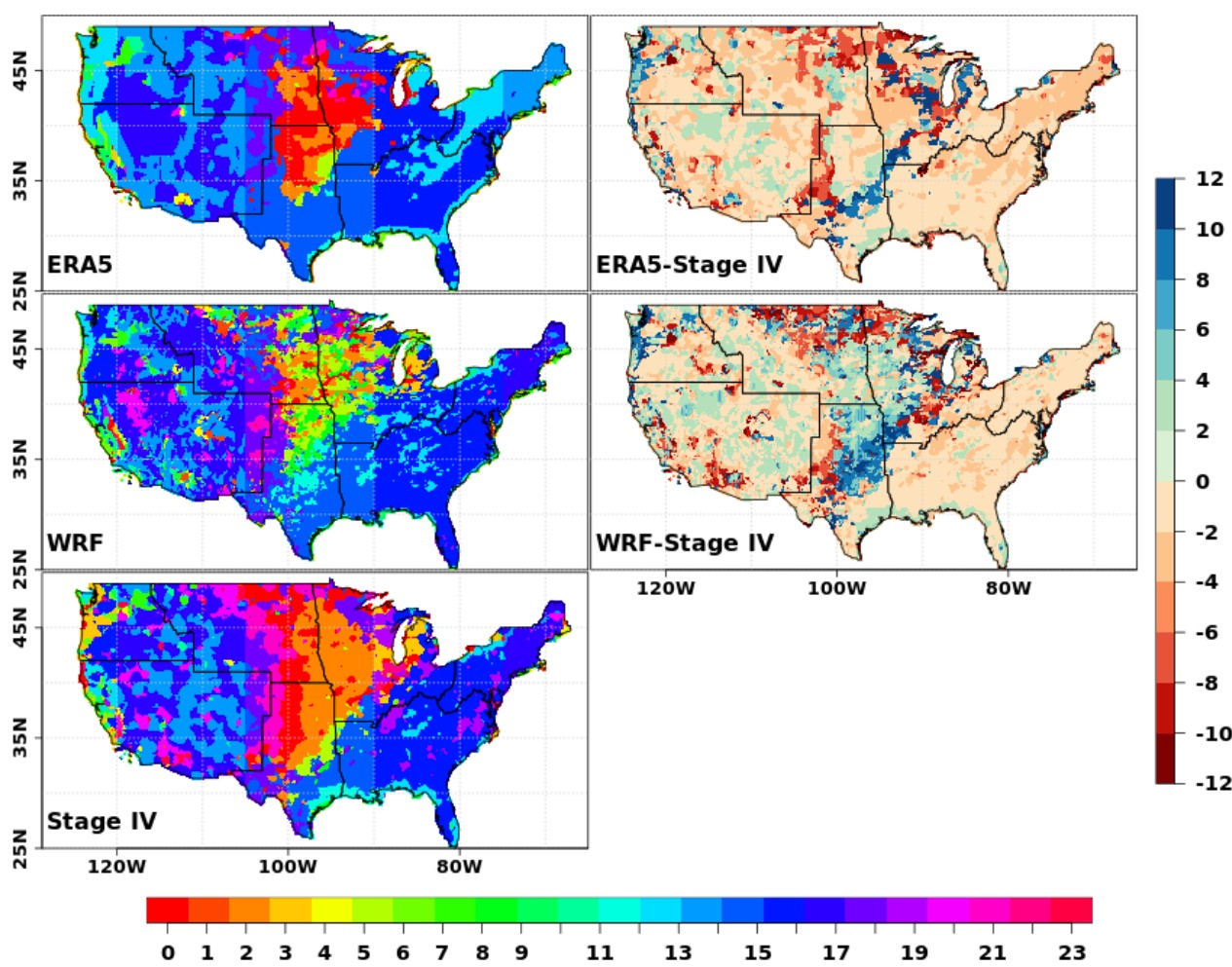

**Figure 2.** Timing of the diurnal precipitation peak (TDPP) in JJA (in units of hours at local solar time) estimated over 2003-2019. The left column shows the timing in each dataset and uses the color scale along the bottom edge of the figure. The right column shows differences in timings of the precipitation peak and uses the color scale along the right edge of the figure.

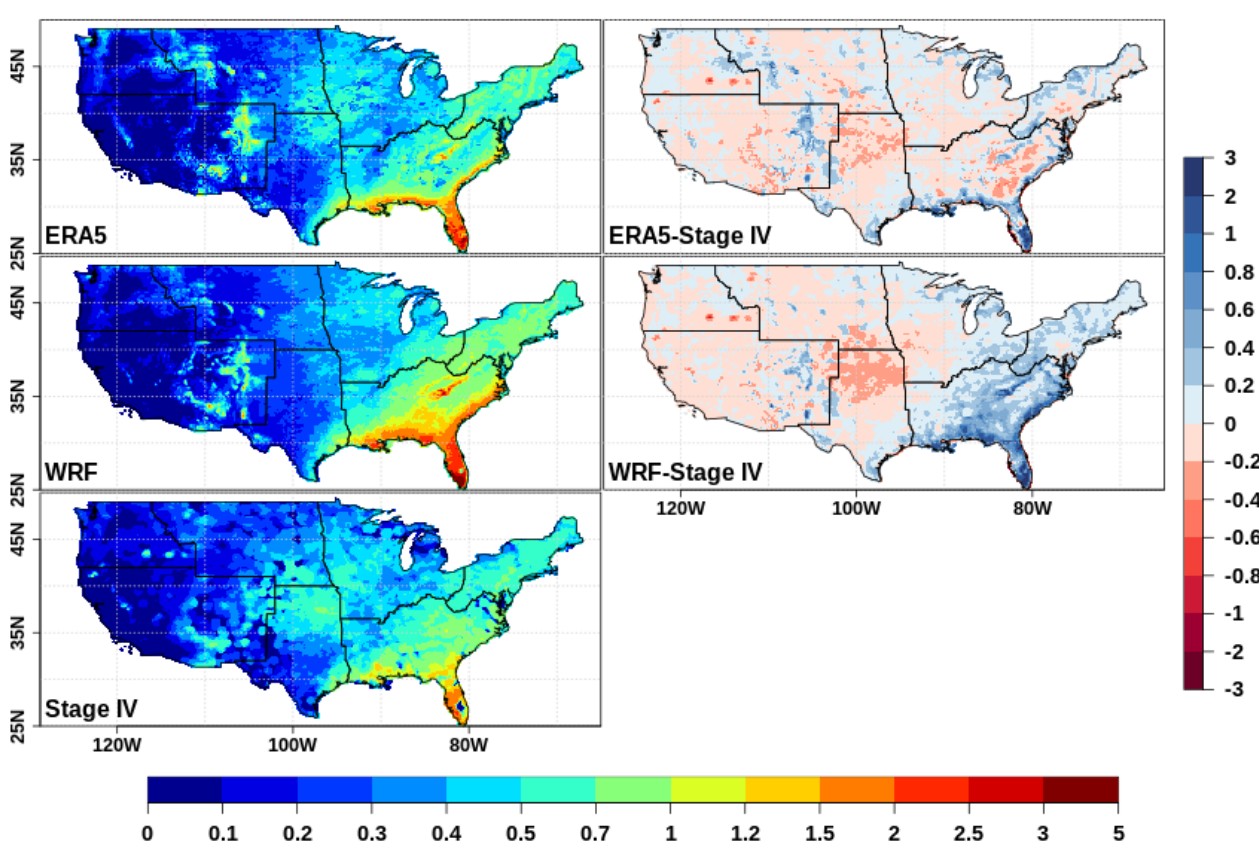

**Figure 3.** Magnitude of the diurnal precipitation peak (MDPP) in JJA estimated over 2003-2019. The left column shows the magnitude in each dataset and uses the color scale along the bottom edge of the figure. The right column shows biases in the magnitude of the precipitation peak and uses the color scale along the right edge of the figure. Units: mm/3hr

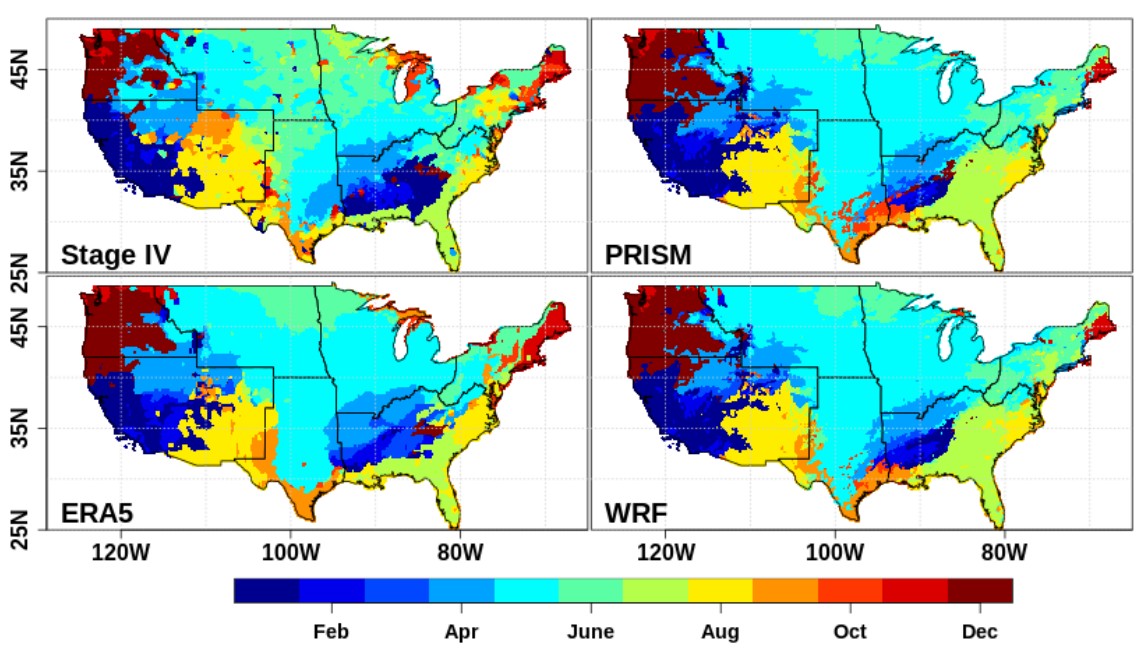

**Figure 4.** Calendar month of the monthly average precipitation peak (TMPP) estimated over 2001-2020 (2003-2019 for Stage IV).

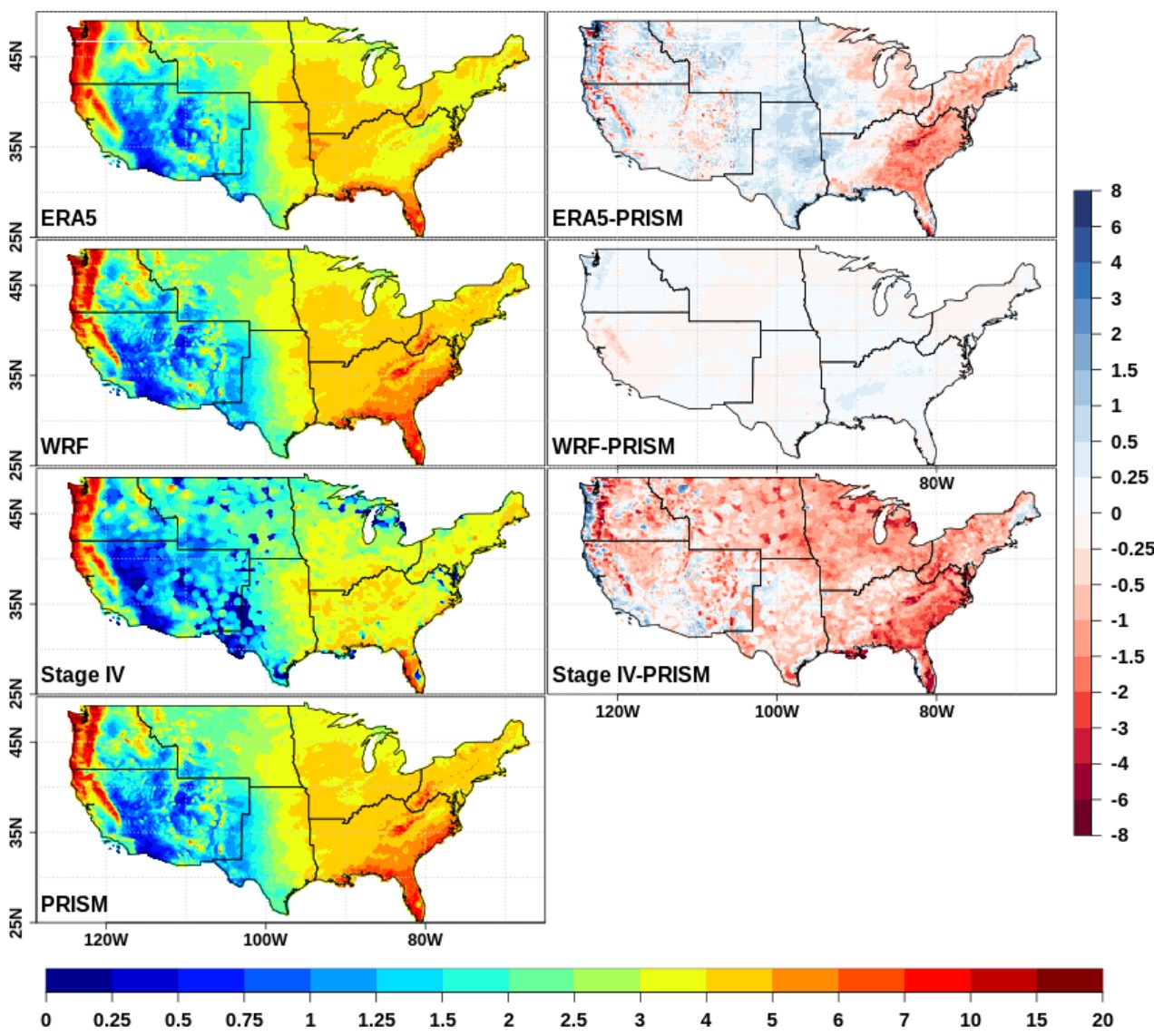

**Figure 5.** Magnitude of the monthly average precipitation peak (MMPP) estimated over 2001-2020 (2003-2019 for Stage IV). The left column shows the magnitude of the peak in each dataset and uses the color scale along the bottom edge of the figure. The right column shows biases in the magnitude and uses the color scale along the right edge of the figure. Units: mm/day.

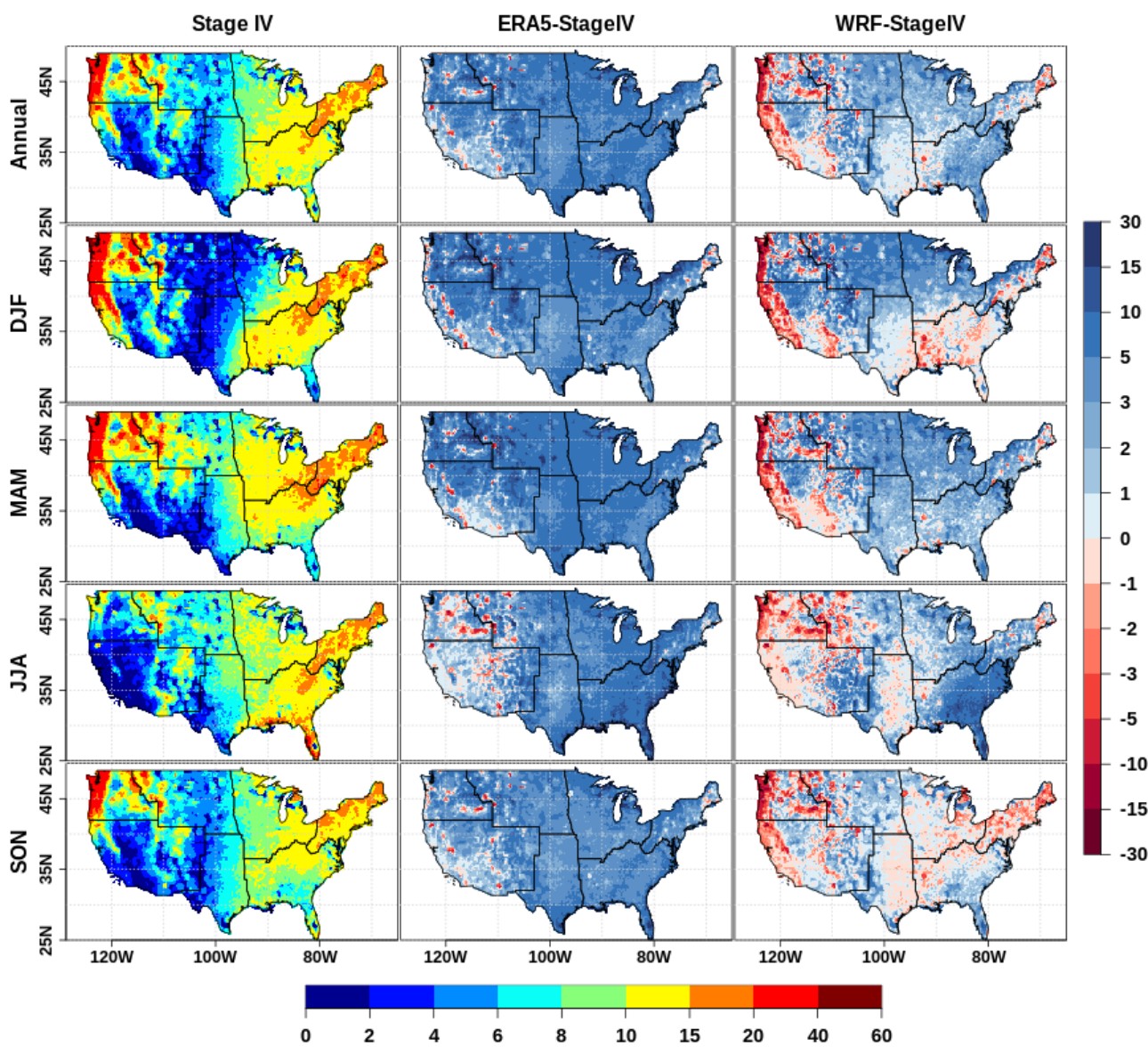

**Figure 6.** 3-hr precipitation frequency (PF3h) estimated over 2003-2019. The left column shows the frequency in Stage IV data and uses the color scale along the bottom of the figure. The right two columns show differences in the precipitation frequency and use the color scale along the right edge of the figure. Units: %.

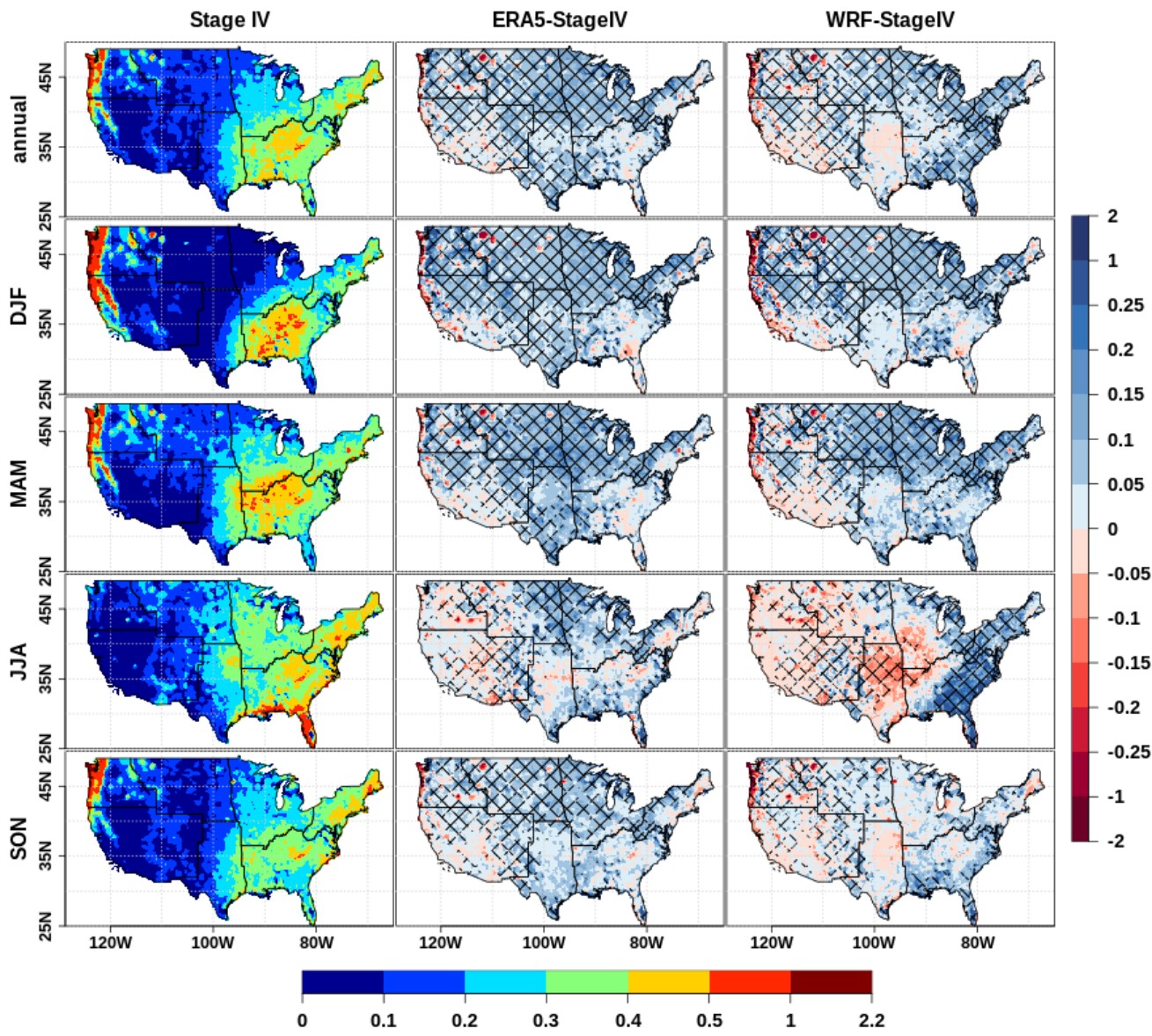

**Figure 7.** 3-hr precipitation mean (Pmean3h) estimated over 2003-2019. The left column shows the mean in Stage IV data and uses the color scale along the bottom of the figure. The right two columns show differences in the mean and use the color scale along the right edge of the figure. Hatching denotes grid points where the differences are found to be significant at the 5% significance level based upon t-test. Units: mm/3hr.

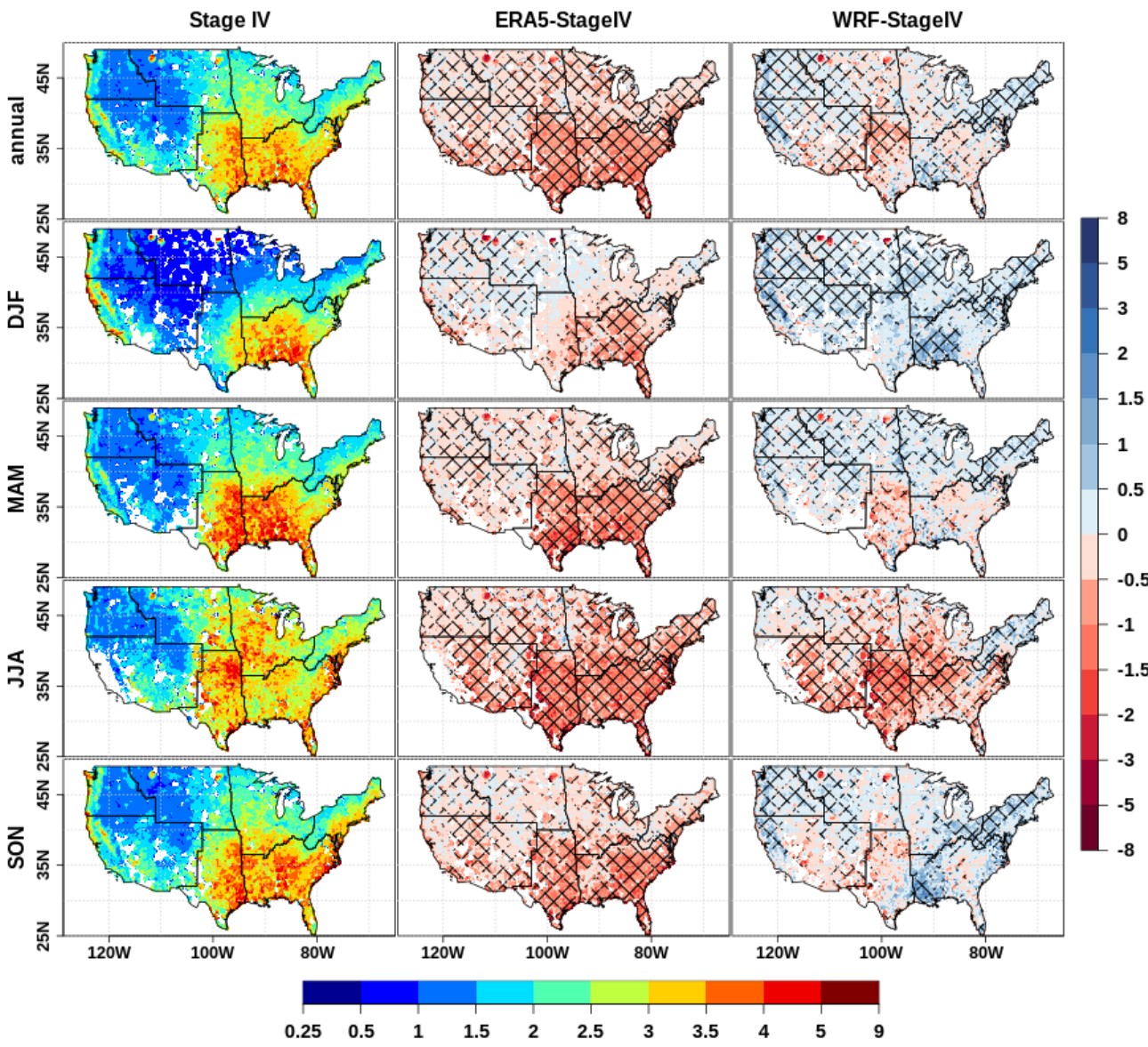

**Figure 8.** 3-hr mean for precipitation greater than 0.25mm (S3hII) estimated over 2003-2019. The left column shows the mean in Stage IV data and uses the color scale along the bottom of the figure. The right two columns show differences in the mean and use the color scale along the right edge of the figure. Hatching denotes grid points where the differences are found to be significant at the 5% significance level based upon t-test. Units: mm/3hr.

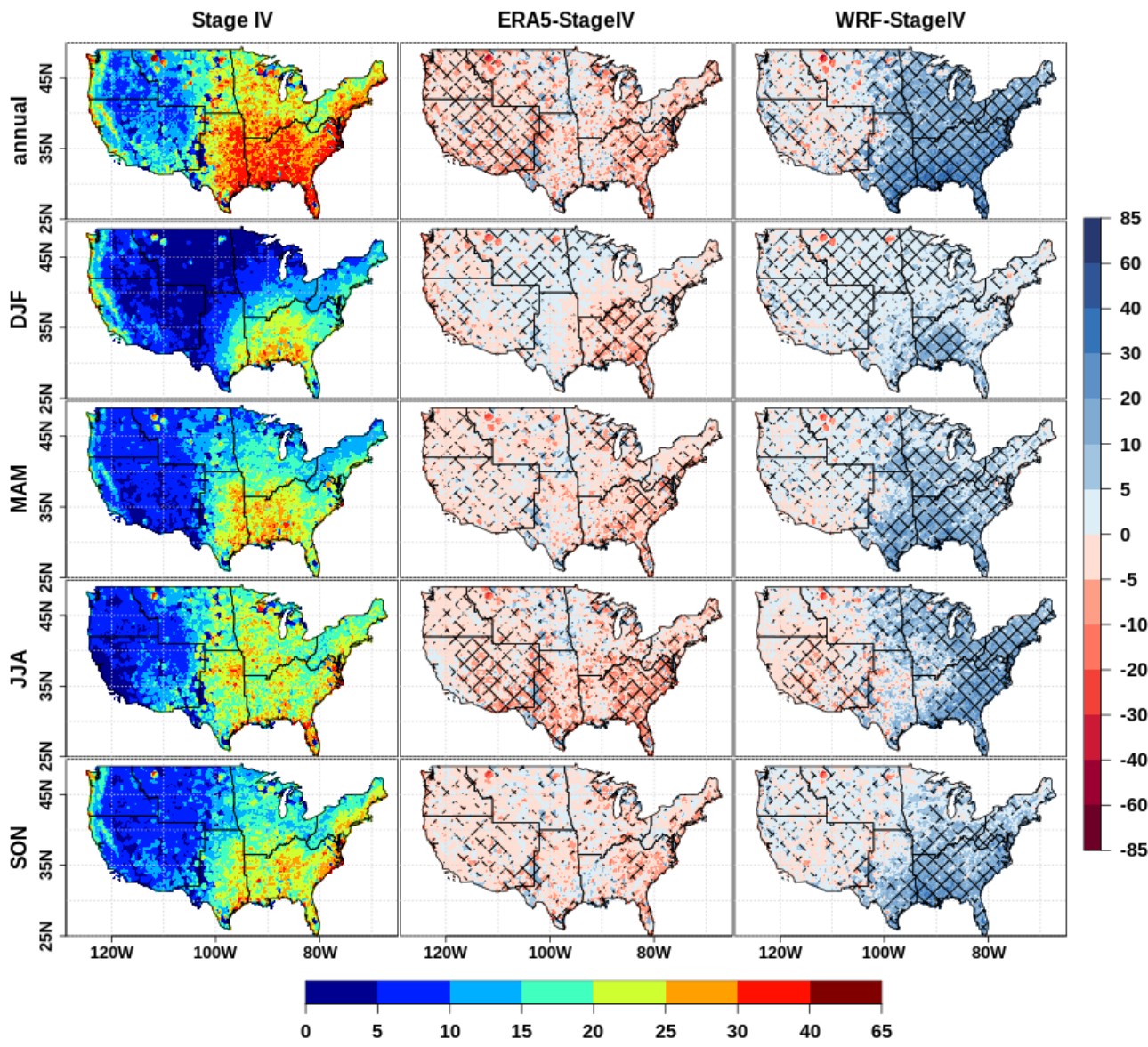

**Figure 9.** 3-hr precipitation maximum (Rx3h) estimated over 2003-2019. The left column shows the mean in Stage IV data and uses the color scale along the bottom of the figure. The right two columns show differences in the mean and use the color scale along the right edge of the figure. Hatching denotes grid points where the differences are found to be significant at the 5% significance level based upon t-test. Units: mm/3hr.

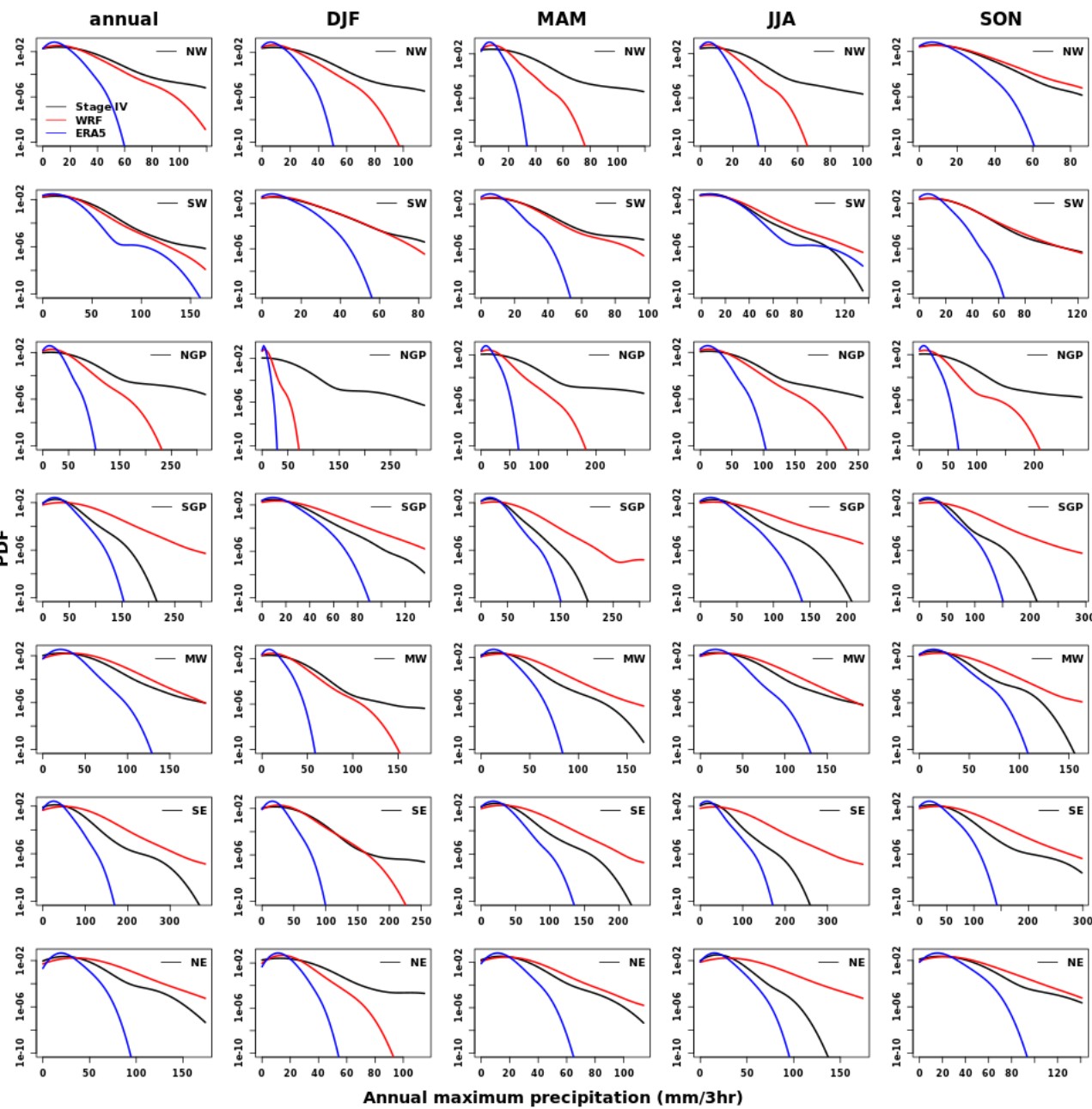

**Figure 10.** Probability density function of 3-hr precipitation annual maximum ((PDF3h) estimated over 2003-2019. The Y-axis is plotted on log-scale.

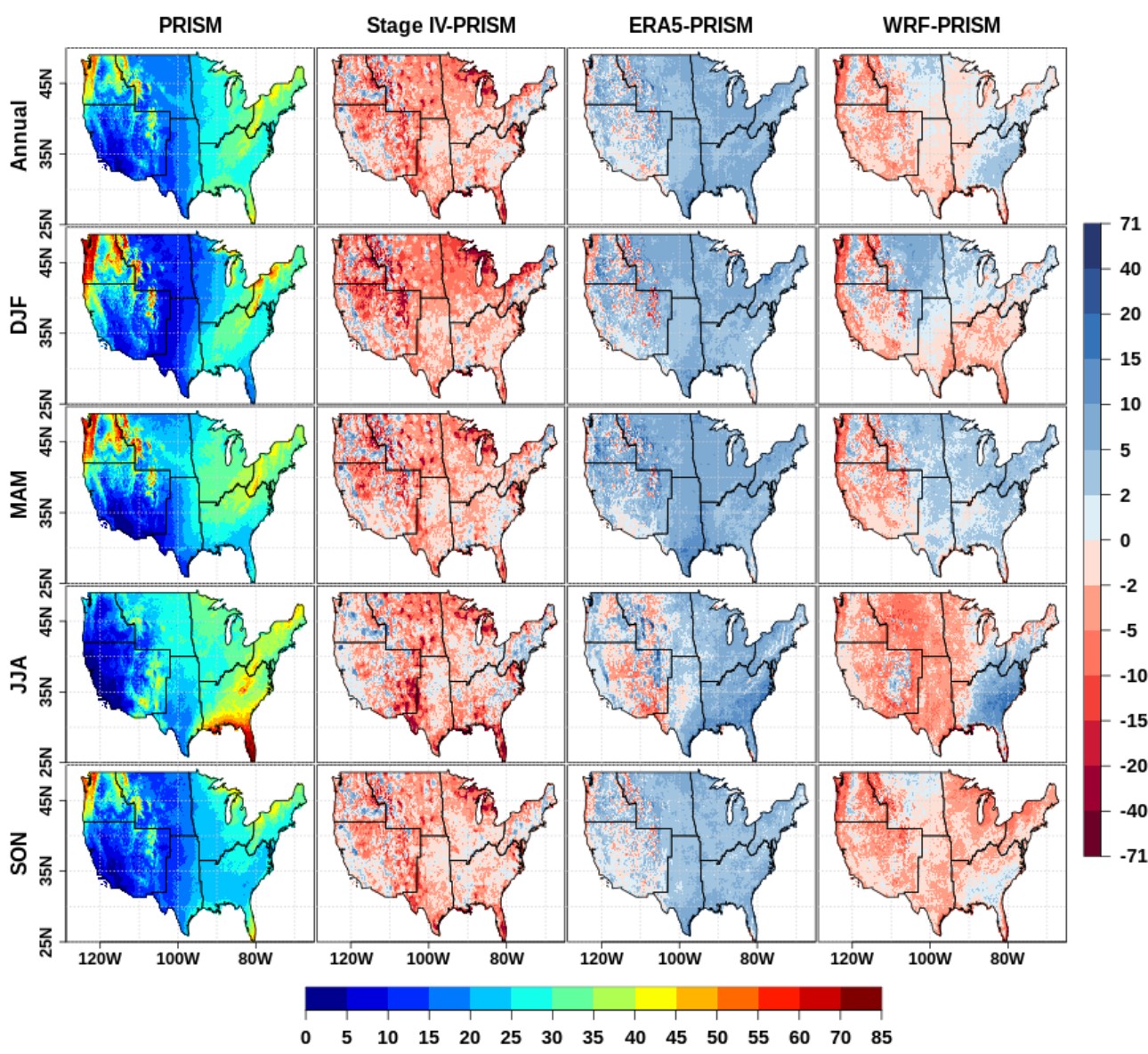

**Figure 11.** As Fig. 6 but for 24-hr precipitation frequency (PF24h) estimated over 2001-2020 (2003-2019 for Stage IV). Units: %

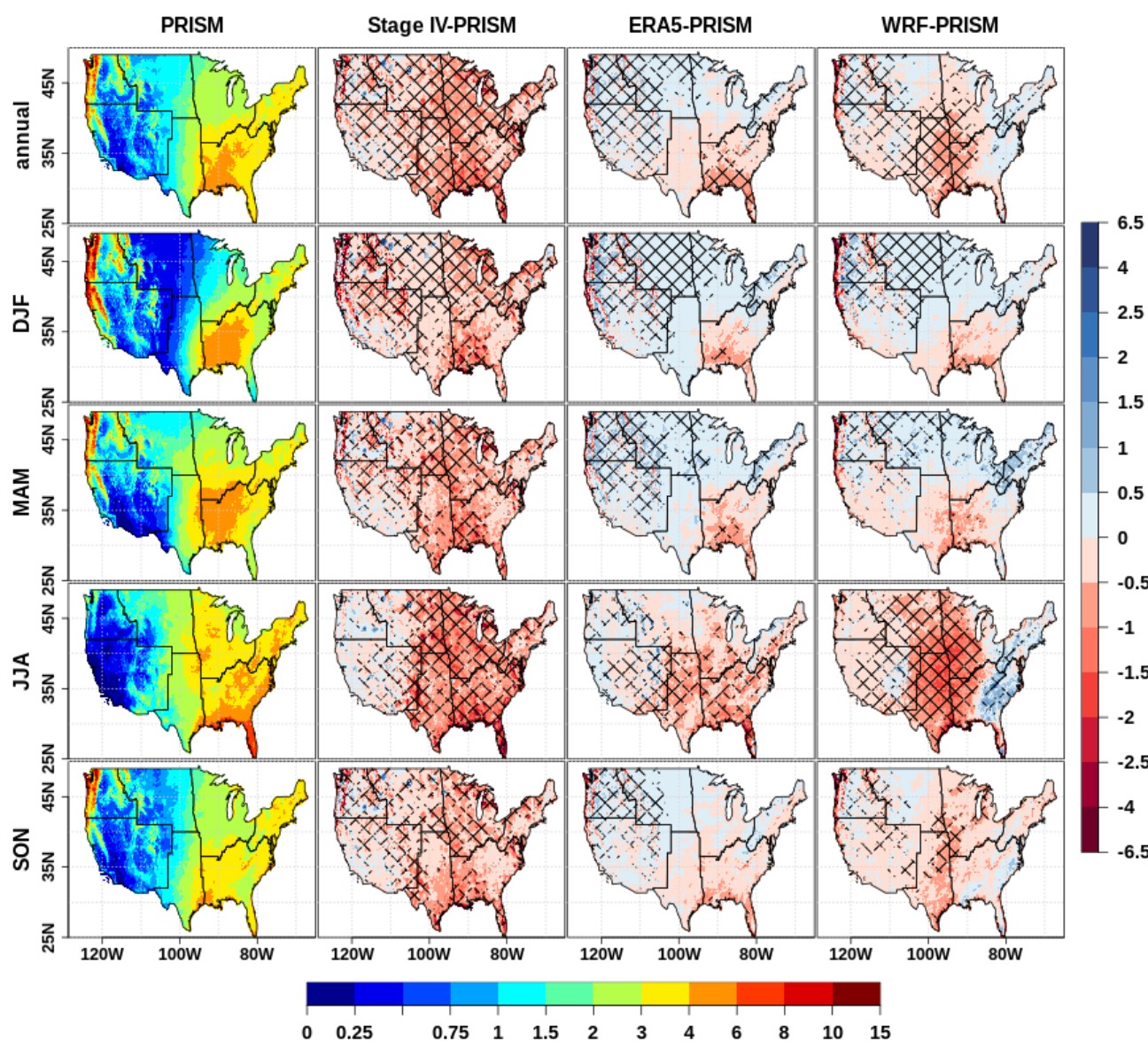

**Figure 12.** As Fig. 7 but for 24-hr precipitation mean (Pmean24h) estimated over 2001-2020 (2003-2019 for Stage IV). Units: mm/day.

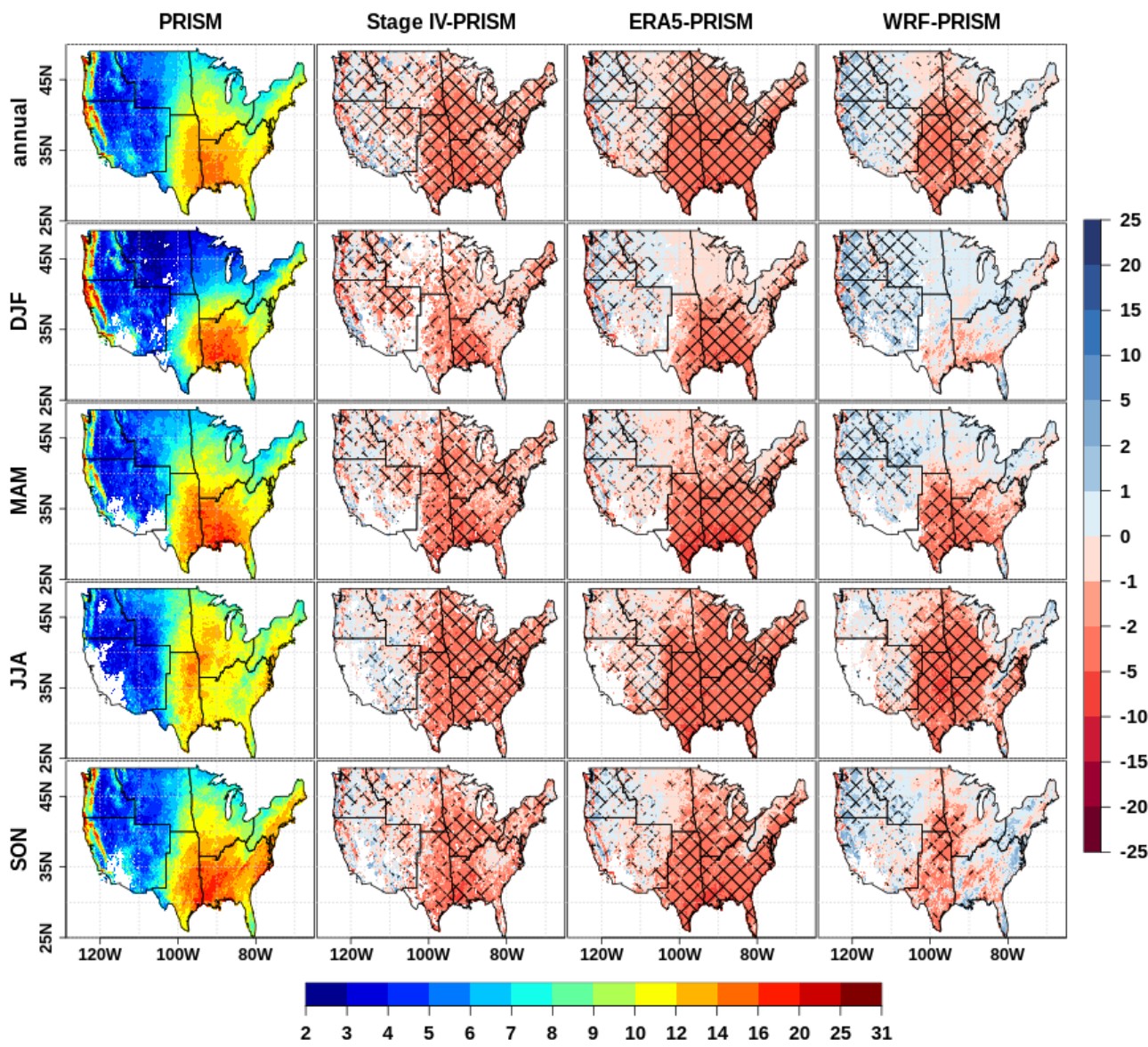

**Figure 13.** As Fig. 7 but for 24-hr mean for precipitation greater than 1mm (SDII) estimated over 2001-2020 (2003-2019 for Stage IV). Units: mm/day.

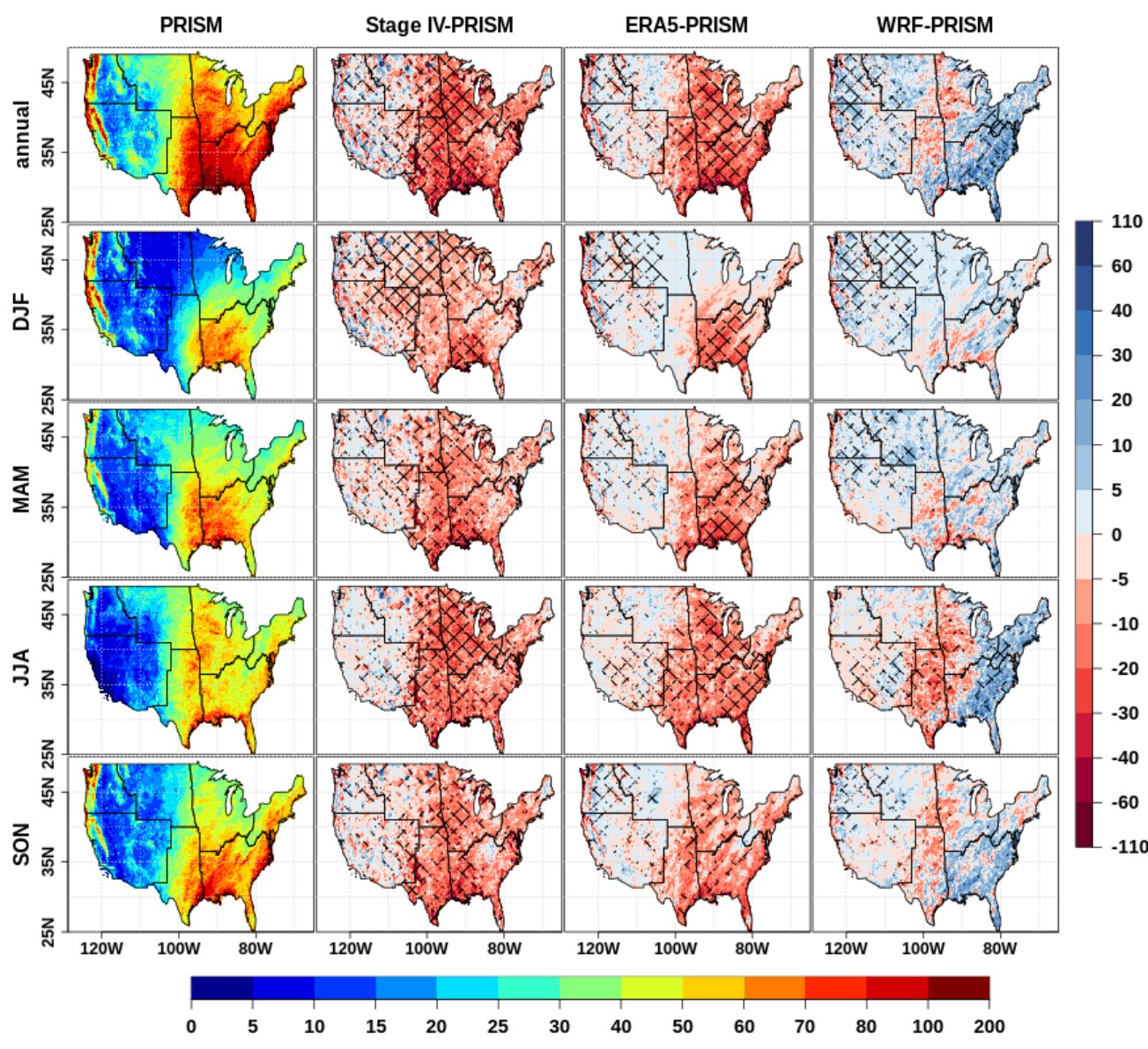

**Figure 14.** As Fig. 9 but for 24-hr precipitation maximum (Rx1day) estimated over 2001-2020 (2003-2019 for Stage IV). Units: mm/day.

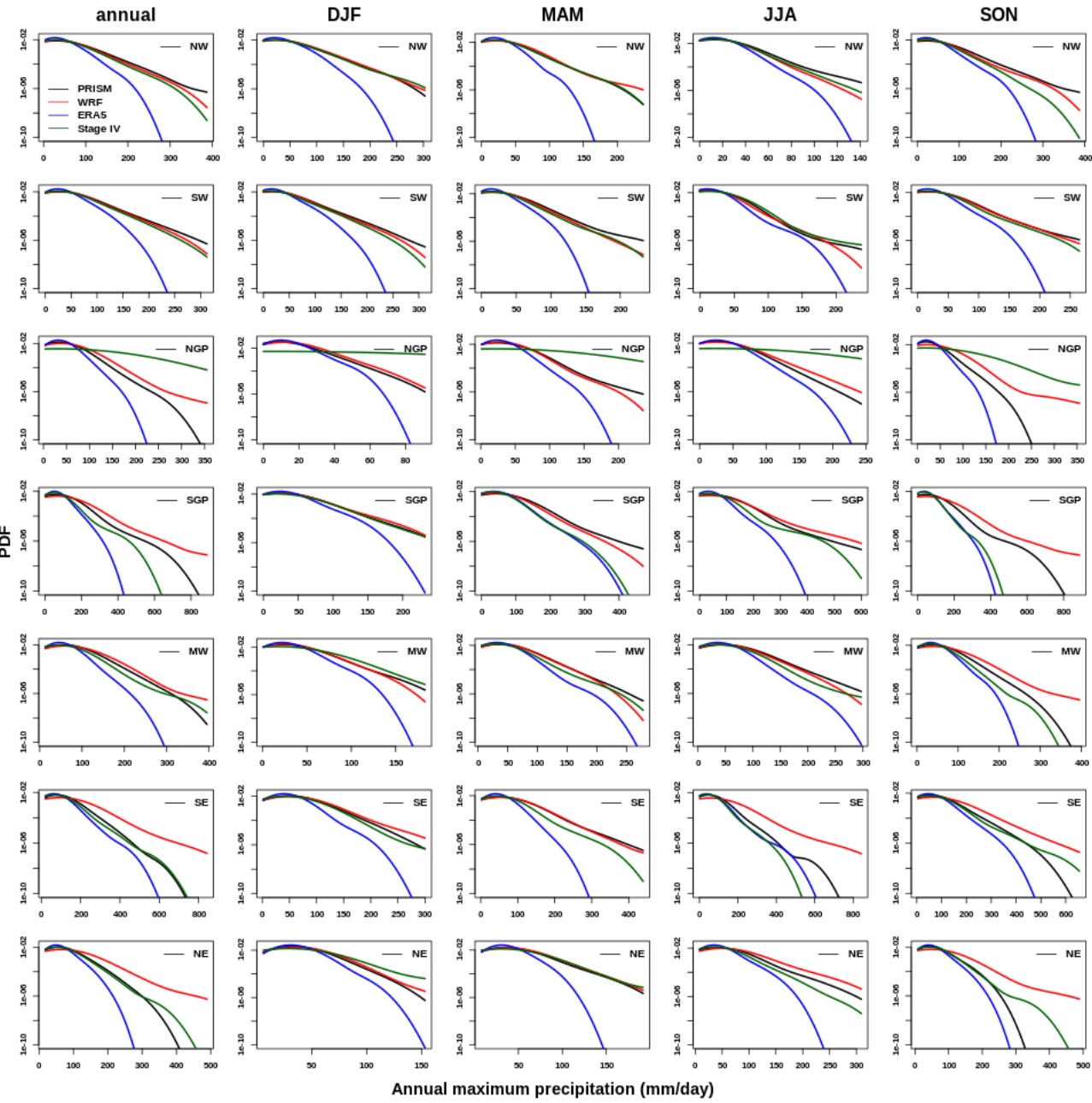

**Figure 15.** As Fig. 10 but for the PDF of 24-hr precipitation maximum (PDF24h) estimated over 2001-2020 (2003-2019 for Stage IV).

**Table 1.** Precipitation metrics analyzed in this study. Please refer to the subsection 2.2 for more details.

| Metrics | Symbol | Description | Unit |
|---------|--------|-------------|------|
| | | 3-hr precipitation | |
| Peak time of the JJA precipitation diurnal cycle | TDPP | Timing of the diurnal precipitation peak in JJA | none |
| Magnitude of the JJA precipitation diurnal cycle | MDPP | Magnitude of the diurnal precipitation peak in JJA | mm/3hr |
| Frequency of 3-hr precipitation | PF3h | Counts of 3-hr precipitation events with magnitude greater than 0.25 mm expressed as a percentage of the total number of 3-hr time steps | % |
| 3-hr precipitation mean | Pmean3h | Climatological mean of 3-hr precipitation including zeros | mm/3hr |
| Wet-3-hr precipitation mean | S3hII | Climatological mean of 3-hr precipitation greater than 0.25 mm | mm/3hr |
| Annual maximum of 3-hr precipitation | Rx3h | Climatological mean of annual maximum of 3-hr precipitation | mm/3hr |
| PDF of the annual maximum of 3-hr precipitation | PDF3h | Probability distribution of the 3-hr annual maximum precipitation | none |
| | | 24-hr precipitation | |
| Peak time of the annual cycle | TMPP | Calander month of the maximum monthly averaged 24-hr precipitation | none |
| Magnitude of the peak of the annual cycle | MMPP | Magnitude of the monthly average precipitation peak | mm/day |
| Frequency of 24-hr precipitation | PF24h | % of total days when 24-hr precipitation is more than 1 mm | % |
| 24-hr precipitation mean | Pmean24h | Climatological mean of 24-hr precipitation including zeros | mm/day |
| Wet-24-hr precipitation mean | SDII | Climatological mean of 24-hr precipitation greater than 1 mm | mm/day |
| Annual maximum of 24-hr precipitation | Rx1day | Climatological mean of annual maximum of 24-hr precipitation | mm/day |
| PDF of the annual maximum of 24-hr precipitation | PDF24h | Probability distribution of the 24-hr annual maximum precipitation | none |

**Table 2.** Regions where ERA5 or WRF precipitation (P) fidelity is subjectively better for 3-hr precipitation. Reference data: Stage IV.

| Precipitation parameter <relevant Figure> | ERA5 | WRF | Comments |
|---|---|---|---|
| Peak time of the JJA precipitation diurnal cycle (TDPP) < Fig. 2 > | The peak occurs earlier over much of the CONUS, especially along the northern boundaries of the Northern Great Plains (NGP) and west of the Great Lakes in the Midwest (MW). | Earlier peak over much of CONUS, better in NE. OEPC (overnight eastward progression of convection) too early in northern NGP, too late in SGP. | Generally larger biases are with OEPC in NGP & parts of: SGP and MW. Both datasets are too late for northern OEPC and too early for southern OEPC. (Note: 12 hrs late = 12 hrs early). Both are too early along NW coast. |
| Magnitude of the JJA precipitation diurnal cycle (MDPP) <Fig. 3> | Better over SE & SGP. Too wet over south FL & Rockies. | Too large over most of SE & less so over NE & eastern MW. Too dry over northern SGP. Better over Rockies. | ERA5 & WRF fine over NW & SW though the magnitude is smaller than elsewhere. |
| Annual frequency of 3-hr P (PF3h) <Fig. 6> | Generally too frequent (>5%) everywhere, less error over southern SW. | Better over NGP, MW, SGP, western SE. SW & NW generally better except not frequent enough along coastal & west-slopes of: NW & SW. | Both datasets too frequent (>5%) over most of: SE, MW, & NGP. |
| Seasonal frequency of 3-hr P (PF3h) <Fig. 6> | Seasons have similar excess as annual except JJA has reduced excess over most of SW & NW. Better at coast & western slopes in DJF, MAM, & SON. | Patterns differ from annual: western SW better during JJA. SGP worse during MAM. SE better in MAM & SON, too frequent during JJA. | ERA5 has frequent P throughout the year, WRF displays seasonal variation. |
| Annual 3-hr precipitation mean (Pmean3h) <Fig. 7> | Generally too wet, except good in SE & southwestern SW. | Best over SGP. Worse over most of SE & NE. Too dry at NW coast. Generally, slightly smaller bias elsewhere. | seasonal biases in WRF over SGP are better except in JJA. |
| Seasonal 3-hr precipitation mean (Pmean3h) <Fig. 7> | MAM: slightly better in NE, SE, SW, & NW coast. JJA: better over SE, SGP, & NE. SON: better in SE, and better along NW coast. | MAM: SGP better. During JJA: SGP & western SE too dry, while eastern SE & all of NE are too wet. SON: better over NGP, MW, and interior SW | DJF similar in both, except NW coast better in ERA5. MAM similar over NGP & MW for both. Though opposite: SON good in both over SGP. |

**Table 2 (Contd . . . ).**

| Precipitation parameter (observation-based dataset) <relevant Figure> | ERA5 | WRF | Comments |
|---|---|---|---|
| Annual wet-3-hr precipitation mean (S3hII) <Fig. 8> | Dry bias, most prominent over SGP and SE | Wet bias over NW,NE and N California. Dry bias over SGP. | Overall smaller biases over NGP, MW & SE. ERA5 shows drizzling bias. |
| Seaonal wet-3-hr precipitation mean (S3hII) <Fig. 8> | DJF: small wet biases over NW, NGP, MW & SW. March-Nov: Strong dry biases over SGP and SE. The dry bias is more widespread and stronger in JJA. | DJF: Generally, wet biases everywhere. MAM: Wet biases over the northern half of the CONUS. Dry over SGP. JJA: Stronger dry bias over SW, SGP, MW & SE. | ERA5 shows drizzling bias. |
| Annual maximum of 3-hr precipitation (Rx3h) <Fig. 9> | Generally too small over whole CONUS, especially eastern SE. | Better over most of NW, NGP, & SW. Much too wet over SGP, MW, SE, & NE. | Most larger values cover SE and eastern SGP. |
| Seasonal maximum of 3-hr precipitation (Rx3h) <Fig. 9> | Generally too small over CONUS though bias least during SON. Only notable area too wet is NGP during DJF. Worst bias during JJA over most of SE & border between SGP & SW. | Generally too wet over MW, NE, & SE though bias least during DJF. Worst biases during JJA too wet over most of MW, NE, & SE while too dry over interior SW. MAM and SON too wet over SE & southern SGP. | WRF shows wet bias in the eastern CONUS, ERA shows dry bias roughly everywhere. |
| Probability density function (PDF) of 3-hr max P (PDF3h) <Fig. 10> | 3-hr max P values are severely underrepresented. | Much better representation. Large underestimation in NGP and overestimation in SGP | NGP and SGP are problematic regions for WRF |

**Table 3.** Regions where ERA5 or WRF precipitation (P) fidelity is subjectively better for 24-hr precipitation. Reference data: PRISM.

| Precipitation parameter (observation-based dataset) <relevant Figure> | ERA5 | WRF | Comments |
|---|---|---|---|
| Peak time of the annual cycle (TMPP) <Fig. 4> | Simulates the spatial pattern except over NE & Gulf regions. | Slightly better over SGP, NE, northern SE, & Great Basin. | NW, SW, NGP, MW, & southeastern SE good in both datasets. |
| Magnitude of the peak of the annual cycle (MMPP) <Fig. 5> | Too dry over most of SE & NE, & eastern NW. Too wet over SGP, much of NGP, & coastal NW. | Generally better over whole CONUS | Both the magnitude and timing of the annual cycle are improved in WRF. |
| Annual 24-hr P frequency (PF24h) <Fig. 11> | Generally better over SW & coastal NW. Generally too frequent over NGP, SGP, MW, NE, & SE. | Generally better over NGP, MW, & NE. Too frequent over eastern SE. Much too infrequent over most of: NW, SW, SGP. | Highest observed over: NE, south FL, coast and mountains of NW & NGP. |
| Seasonal 24-hr P frequency (PF24h) <Fig. 11> | DJF better along coastal NW & most of SW. DJF, MAM, & SON: too frequent over NGP, MW, NE, SE, SGP, and most of: SW & NW. | DJF is better over most of MW, NE, & SGP. Coastal & mountainous: NW & SW are generally too infrequent during DJF, MAM, & SON. SE too infrequent during DJF but other seasons too frequent. JJA: much too infrequent over all but opposite bias over parts of SE & NE. SON: too infrequent over most of CONUS | Both datasets too infrequent along NW coast during DJF & SON, though ERA5 better there. Both too infrequent during JJA over most of SW. |
| Annual 24-hr precipitation mean (Pmean24h) <Fig. 12> | Worse biases (dry) over southern SE. | Worse biases (dry) over western SE and most of SGP. | observed peak values over western SE and coastal NW. Datasets generally similar except SE, SGP, & coastal NW where WRF has greater dry bias. |
| Seasonal 24-hr precipitation mean (Pmean24h) <Fig. 12> | MAM & SON biases generally similar to DJF. JJA has largest biases (dry) covering all of: SGP & SE & much of: MW & NE. JJA & SON better over NGP, MW, SGP, & western SE. | MAM bias similar to DJF. Largest bias (dry) is during JJA and covers SGP, most of: MW & NGP, and western SE. SON: coastal SE is better. | DJF: similar in both datasets with greater bias along coast of NW, & Gulf of Mexico coast of SE. Largest seasonal values are at coastal NW during DJF; ERA5 captures this better. Secondary maximum during JJA covers FL and SE Gulf of Mexico coast; both datasets underestimate these larger values. |

**Table 3 (Contd . . . ).**

| Precipitation parameter (observation-based dataset) <relevant Figure> | ERA5 | WRF | Comments |
|---|---|---|---|
| Annual wet 24-hr precipitation mean (SDII) <Fig. 13> | Strong dry biases over eastern CONUS. | Worse biases (dry) over western SE, SGP & southern MW+NGP regions wet biases over NW and SW. | Smaller biases over NGP, MW, NE & SE. |
| Seasonal wet 24-hr precipitation mean (SDII) <Fig. 13> | DJF & MAM: Strong dry biases over SE and eastern SGP. Small wet bias over NW. JJA & SON: Strong dry bias over eatsren half of the CONUS | DJF: Generally, small wet biases except over SGP and SE. JJA: Strong dry bias over the Great Plains and SE. SON: Dry bias over the SGP, small wet bias over NW. | Biases are typically smaller in magnitude in WRF in DJF and SON. |
| Annual 24-hr max P (Rx1day) <Fig. 14> | General dry bias over SGP, MW, SE, & NE. Slightly better over NW & SW. | Wet bias over most of SE, NE, eastern MW, & parts of: SW, NGP, & NW. | Datasets do well over NW, NGP, & SW. They have opposite biases over most of SE, NE, MW, & southern SGP. |
| Seasonal 24-hr max P (Rx1day) <Fig. 14> | DJF: dry bias mainly in SE. MAM, JJA, & SON: dry bias across SGP, SE, MW, & NE. | DJF: better over SE. MAM & SON: better over most of: SGP, SE, MW, NE. JJA: wet bias over: NE & eastern and southern SE. | Performance similar over SW, NW, & NGP. Both have large dry bias over SGP during JJA. Bias generally smaller over SW, NW, and NGP, but so are the observed means. |
| Probability density function (PDF) of 24-hr max P (PDF24h) <Fig. 15> | 24-hr max P values are severely underrepresented. | Much better representation. | 24-hr PDF representation is better than the 3-hr PDF in WRF. |