# Peer review of "Assessment of WRF (v 4.2.1) dynamically downscaled precipitation on subdaily and daily timescales over CONUS"

_EGUsphere, 2022_

## Author Comment (AC1)

Manuscript ID: egusphere-2022-1382

Response to reviewer # 1

General comments: In general, this study evaluated sub-daily and daily precipitation data from a WRF simulation over CONUS against NCEP and PRISM datasets. This paper is well-written and logically flows well. The findings and caveats in WRF simulations are comparable to earlier studies. I have one major comment and several minor comments and hope the authors can address them.

Response: We sincerely thank the reviewer for the positive review. We have thoroughly revised the manuscript in response to your comments. Our responses to your comments are given below. In addition, we have also included a file (Srivastava_marked-difference) with the revised submission, indicating changes in the manuscript in blue color.

Comment #1: The second paragraph in Summary and Discussion is way too dense and hard to read. I would recommend the authors split it into two or even three paragraphs and reorganize it to increase readability.

Response: We have reduced the size of the second paragraph in the Summary and Discussion section. We have also reorganized the section in response to another reviewer's comments. Please refer to the revised Summary and Discussion section in lines 308-372 of the revised manuscript.

**Specific comments:**

**Comment #2. L65: Please simply use the regional climate model instead of RCM since this is the only time that the abbreviation was used in this manuscript.**

**Response:** "regional climate model" is used instead of RCM in line 65 of the revised manuscript.

**Comment #3. Figure 1: can the authors please 1) highlight the domain for this study, 2) add a topography layer, and 3) add the NCA region boundaries and names to this map for better illustration purposes?**

**Response:** The revised Figure 1 includes the domain, topography, and NCA region boundaries.

**Comment #4. L163: The selection of the 0.25mm threshold seems random. Please justify it.**

**Response:** We have added the following text to justify 0.25mm and 1mm thresholds used in the study. Please refer to lines 110-118 of the revised manuscript:

L110-118: In this study, we estimate precipitation metrics that characterize the frequency, total amount, intensity, and timing of the mean and extreme precipitation. The metrics are summarized in Table 1. We calculate the mean precipitation amount for 3- and 24-hr durations using all precipitation values, including zeros. We use 0.25mm and 1mm thresholds for estimating the frequency and mean precipitation during wet 3-hr and 24-hr

periods, respectively. We use these thresholds to minimize the effect of excessive drizzle being present in regional climate models and reanalyses (e.g., Frei et al., 2006; Rajczak et al., 2013), and also to account for observational constraints (Schär et al., 2016). The differences between the mean precipitation amount and the mean wet-3- hr/ wet-24-hr precipitation highlight the biases that result from excessive drizzle in the dataset. The precipitation thresholds in the study are consistent with those in previous studies (e.g., Rajczak et al., 2013; Rajczak and Schär, 2017; Xiao et al., 2018; Kooperman et al., 2022)

**Comment #5. L193-198: can the authors please explain why WRF improves less on capturing extreme precipitation values in the NGP and SGP regions?**

**Response:** Please refer to the following text in lines 233-236 of the revised manuscript.

L 233-236: A detailed investigation of biases in WRF is out of the scope of this paper, but we suspect that WRF biases in the Great Plains may be attributed to underestimated MCS frequencies (Prein et al., 2020), imperfect cumulus parameterization scheme and biases in the representation of intensity, location, and diurnal cycle of the low-level jet in 12-km WRF simulation (Lee et al., 2017).

**Comment #6. L201: Please justify the selection of 1mm and add references if any.**

**Response:** Please refer to our response to your comment #4 above.

**Comment #7. L233: Should it be "For example, it shows wet biases during winter and spring, but a mix of wet and dry biases during summer and fall?"**

**Response:** Thank you for the comment. We use the following text in lines 297-298 of the revised manuscript:

L297-298: For example, it shows wet biases during winter and spring but a mix of wet and dry biases (SGP and MW) during summer and fall.

---

## Author Comment (AC2)

**Response to reviewer # 2**

General comments: Review of "Assessment of WRF (v 4.2.1) dynamically downscaled precipitation on subdaily and daily timescales over CONUS" by Srivastava et al, 2023. Submitted to GMD.

This paper compares the results from a 12km WRF dynamically downscaled climate experiment driven with ERA5 boundary conditions with ERA5 itself, Stage-IV precipitation (for 3-hourly fields) and PRISM (for daily fields). This paper examines the diurnal cycle, seasonal cycle, precipitation frequency, and precipitation intensity. This paper was well written and easy to follow. This paper should be accepted for publication after minor revisions suggested below.

**Response:** We sincerely thank the reviewer for the positive and helpful comments. We have thoroughly revised the manuscript in response to your comments. Our responses to your comments are given below. In addition, we have also included a file (Srivastava_marked-difference) with the revised submission, indicating changes in the manuscript in blue color.

Overall Comments:

Comment #1. I suggest the authors also create a daily product from the Stage-IV 3-hourly product to compare with the PRISM data, ERA5, and WRF-ERA5 or provide a discussion of how Stage-IV data compares with PRISM from other papers.

**Response:** Thank you for your comment. We have now included daily

precipitation analysis of Stage IV data in the revised manuscript. Section 3.4 of the revised manuscript now includes a discussion on the performance of Stage IV against PRISM data. Please refer to the following lines of the revised manuscript:

L244-245: For 24-hr precipitation analysis, we use PRISM as reference data. We also evaluate 24-hr precipitation in Stage IV against PRISM to quantify observational uncertainty.

L247-252: Stage IV consistently underestimates (in comparison to PRISM) the precipitation frequency over most of the CONUS. The largest biases in Stage IV precipitation frequency are observed over the NGP in winter and SW throughout the year (Supplementary Fig. S7). The underrepresented precipitation frequency in Stage IV may be related to its difficulty in detecting light and frozen precipitation across the CONUS and, most notably, in the western US, because the precipitation processing system in Stage IV does not distinguish between liquid and frozen hydrometeor types (Smalley et al., 2014)..

L262-266: Stage IV shows dry bias as compared to PRISM over most of the CONUS in all seasons, except that it shows wet biases over sporadic locations in NW and SW regions. The corresponding percent bias in Pmean24h (Supplementary Fig. S8) indicates large Stage IV relative dry biases in the western CONUS (NGP, NW and SW) in DJF, possibly related to its inability to detect freezing and light precipitation events, as discussed in the previous subsection.

L291-294: As for the other metrics, Stage IV underestimates Rx1day over the eastern half of the CONUS. The dry bias is most pronounced ($\sim$ 20%) over the Great Plains and MW during summer and over the NGP and northeastern parts of SW ($> 50\%$) during winter (Supplementary Material Fig. S10). On the other hand, Rx1day values in Stage IV are very well represented over NW and SW in all seasons except winter.

L301-302: Stage IV represents well the PDF24h over NW and SW. However, it does show problems in capturing the PDF24h over the NGP throughout the year.

**Comment #2. It is unclear what the authors mean by "operational purposes" in their paper. Perhaps WRF climate simulations would be used to help inform stakeholders about future precipitation/water availability- but such climate simulations that these WRF simulations represent are not "operational" in any traditional sense.**

**Response:** We have revised the text in the abstract for clarity. The revised text is as follows.

L17: Consequently, if used as input data for domain-specific models, we suggest moderate bias-correction be applied to the dynamically downscaled product.

**Comment #3. It was unclear to me if you included the zeros in your mean precipitation plots for 3-hrly precipitation and daily precipitation means. That should be made clear because that does**

**influence if the plots are really "average precipitation intensity" or just "average precipitation" - you make some statements about precipitation intensity vs frequency, but if you average with the zeros you don't really capture how much it rains when it rains.**

**Response:** We thank the reviewer for this helpful comment. In the revised manuscript, we have used 3-hr precipitation (Pmean3h) and daily precipitation means (Pmean24h) that use zero precipitation values. We have also used wet-3-hr precipitation mean (S3hII) and wet-24-hr precipitation mean (SDII), which are the means of precipitation values greater than 0.25mm and 1mm, respectively. Please refer to Table 1 of the revised manuscript for the definition of these indices. The revised manuscript includes a discussion on S3hII and SDII as follows.

L110-118: In this study, we estimate precipitation metrics that characterize the frequency, total amount, intensity, and timing of the mean and extreme precipitation. The metrics are summarized in Table 1. We calculate the mean precipitation amount for 3- and 24-hr durations using all precipitation values, including zeros. We use 0.25mm and 1mm thresholds for estimating the frequency and mean precipitation during wet 3-hr and 24-hr periods, respectively. We use these thresholds to minimize the effect of excessive drizzle being present in regional climate models and reanalyses (e.g., Frei et al., 2006; Rajczak et al., 2013), and also to account for observational constraints (Schär et al., 2016). The differences between the mean precipitation amount and the mean wet-3- hr/ wet-24-hr precipitation highlight the

biases that result from excessive drizzle in the dataset. The precipitation thresholds in the study are consistent with those in previous studies (e.g., Rajczak et al., 2013; Rajczak and Schär, 2017; Xiao et al., 2018; Kooperman et al., 2022).

L219-226: Fig. 8 shows the 3-hr mean for precipitation greater than 0.25mm/3hr (S3hII). As shown for Stage IV, mean S3hII values are generally higher than Pmean3h across the CONUS. The highest S3hII values are observed over the SE and SGP regions, suggesting that 3-hr precipitation in these regions is dominated by drizzling precipitation ($< 0.25$ mm). Notably, except for parts of NGP, NW, and SW regions in DJF, ERA5 underestimates the mean S3hII over most of the CONUS in all seasons. This ERA5 bias, together with those shown in Figs. 6 and 7 suggest that ERA5 suffers from drizzling effect, causing it to precipitate more frequently but in lesser amounts when it rains. In contrast to ERA5, WRF simulates more S3hII values across the CONUS in DJF and less in JJA. Notably, the absolute S3hII biases in WRF are generally lower than those in ERA5 in most of the seasons and regions.

L284-290: The 24-hr mean wet-day precipitation (SDII) is shown in Fig. 13. As for the biases in Pmean24h (Fig. 12), Stage IV underestimates SDII almost everywhere, but more prominently over the eastern half of the CONUS in all seasons. ERA5 underestimates SDII over the eastern half of the CONUS (parts of NGP, MW, SGP, and NE) across the year. The dry SDII biases, together with the overestimated frequency and mean precipitation in winter and spring over NGP, MW, and SGP, suggest that ERA5 has too-little-and-too-frequent precipitation bias. WRF exhibits wet SDII biases over most of the CONUS in DJF, except in a few places over the SGP and SE. On the other hand, it shows strong dry biases over the SGP and SE during spring and over the SGP, MW and SE during summer.

**Comment #4. The utility of the paper to end-users of this data would be improved if discussions of how much bias is tolerable for a model to be useful and if the WRF/ERA5 data are within that range.**

**Response:** The revised text in lines 355-366 highlights this point.

L 355-366: A related question is how much bias is acceptable in a climate model. The acceptable level of biases really depends on the application of the climate data. Although the data could be used directly in analysis, we expect a large portion of users will use the data to force other models. In that case, tolerance for biases depends on the type, scope, and scale of the downstream modeling frameworks. Nonetheless, the question is hard to answer quantitatively given that a large uncertainty exists even among observational datasets (e.g., Srivastava et al., 2020, 2022). Still, one can qualitatively assess the model's performance by comparing it with other models or observational datasets. We assessed the observational uncertainty in 24-hr precipitation representation by comparing precipitation characteristics between PRISM and Stage IV in 24-hr precipitation analysis. We found that biases in WRF are generally smaller in magnitude than in Stage IV. For example, annual

24-hr precipitation frequency (PF24h) is better simulated in WRF than in Stage IV, and biases in the magnitude of monthly average precipitation peak (MMPP) are much smaller in WRF than in Stage IV. Similarly, WRF shows comparable (e.g., DJF PDF24h in NW and SW) or even better (e.g., NGP in all seasons) simulation of Rx1day PDF (PDF24h) than Stage IV. These analyses suggest that WRF reasonably simulates the observed precipitation characteristics across the CONUS.

**Comment #5. Percent biases (maybe added to supplemental) could be really informative.**

**Response:** Thank you for the valuable comment. We have included percent bias figures in the Supplementary Material.

**Comment #6. Mean plots are often too "blue". I suggest using a more-non linear scale to highlight more differences.**

**Response:** We now use a more nonlinear color scale in figures to highlight differences.

**Minor comments:**

**Comment #7. L28: update to "processes facilitates the study of future changes"**

**Response:** We have revised the text in lines 27-28 of the revised manuscript.

**Comment #8. L34-35: sentence a little unclear as written perhaps: "biases are generally not consistent across the variables, regions, and seasons of interest" or just "biases vary with variable, region, and season.**

**Response:** We have rewritten the text in lines 34-35 to bring clarity.

**Comment #9. L46: no the needed in front of historical so "we evaluate historical"**

**Response:** Fixed.

**Comment #10. L83: How might the inclusion of urban surfaces influence precipitation in these simulations?**

**Response:** The text in lines 85-91 includes the discussion on urbanization.

L85-91: Studies suggest that urbanization can enhance or suppress precipitation over different regions, situations, and urbanization phases. Some examples are: Wang et al. (2015) show that urban warming during the early urbanization phase promotes increased sensible heat flux, enhanced convergence, and vertical motion, leading to urban modification of rainfall. Li et al. (2022) find that urbanization suppresses summer precipitation from mesoscale convective systems, isolated deep convection, and non-convective systems in the Mid-Atlantic region east of the Rocky Mountains. Georgescu et al. (2021) report that physical growth of the built environment can either enhance or suppress extreme precipitation across CONUS metropolitan regions.

**Comment #11. L89-94: I think this would read better if there was no "the" in front of Stage VI.**

**Response:** Corrected.

**Comment #11. Section 2.1.1 - How well do Stage IV and**

PRISM perform in mountains?

**Response:** We have not especially analyzed how Stage IV and PRISM perform in mountains, but have added the following text to show their performance in western US in general.

L248-252: The largest biases in Stage IV precipitation frequency are observed over the NGP in winter and SW throughout the year (Supplementary Fig. S7). The underrepresented precipitation frequency in Stage IV may be related to its difficulty in detecting light and frozen precipitation across the CONUS and, most notably, in the western US, because the precipitation processing system in Stage IV does not distinguish between liquid and frozen hydrometeor types (Smalley et al., 2014).

**Comment #12. Section 2.1.2 - I think a small statement about what "solar noon" means would help contextualize this work for people who do not work with diurnal cycle data.**

**Response:** Following line has been added to the revised manuscript.

L122-123: 12 noon LST is the time when the Sun is highest in the sky at a location.

**Comment #13. L110: What do you mean "variability is assumed to be the same in 20 year period"? Is this saying you picked a 20-year period similar to the SageIV for the PRISM data because then internal variability wouldn't play a role? I think this needs to be expanded slightly for clarity.**

**Response:** We revised the text to bring clarity.

L128: Second, any variability arising from the trend may be assumed to be insignificant in the 20-year record.

**Comment #14. L138: You say here ERA5 against two different reference datasets - but so far you have only done Stage IV so that is confusing**

**Response:** Here we meant the reference datasets used in our study and the one used by Watters et al (2021). We ahve revised the text to bring clarity.

L156-157: The differing performance of ERA5 against the two different observational datasets (as noted in Watters et al. (2021) and our study) also points to uncertainties arising due to differences in reference datasets.

**Comment #15. L144: You mention MCS and the Great Plains, but why is there too much precip in the southeast?**

**Response:** We have included the following text in the revised manuscript.

L162-168: The wet MDPP bias in WRF over the SE is also observed in previous WRF-based studies (e.g., Wang and Kotamarthi, 2014; Scaff et al., 2020). Sun and Bi (2019) showed that the WRF simulation with the Tiedke cumulus parameterization scheme exhibits an earlier and stronger diurnal cycle than the observed over land regions between 25°S and 25°N in boreal summer. As the convective scheme is the most crucial model component in capturing the diurnal cycle of precipitation (Shin et al., 2007); and precipitation from cumulus parameterization schemes dominates over the SE CONUS (Iguchi et al., 2017), we suspect that cumulus parameterization in the current

WRF simulation may be responsible for the wet bias over the SE region.

**comment #16. Section 3.2 - I am curious how stage IV and PRISM compare (see note at top).**

**Response:** We have now included the text comparing Stage IV and PRISM in the revised manuscript. Please refer to the following text.

L186-189: Stage IV does capture the spatial pattern of the referenced precipitation magnitude; it exhibits underestimated precipitation (dry bias) of 20% or more almost everywhere across the CONUS. The largest percent biases exist over the SE and SW regions (Fig. 5 and Supplementary Fig. S2).

L247-252: Stage IV consistently underestimates (in comparison to PRISM) the precipitation frequency over most of the CONUS. The largest biases in Stage IV precipitation frequency are observed over the NGP in winter and SW throughout the year (Supplementary Fig. S7). The underrepresented precipitation frequency in Stage IV may be related to its difficulty in detecting light and frozen precipitation across the CONUS and, most notably, in the western US, because the precipitation processing system in Stage IV does not distinguish between liquid and frozen hydrometeor types (Smalley et al., 2014)

L262-266: Biases in 24-hr precipitation mean (Pmean24h) are shown in Fig. 12. Stage IV shows dry bias as compared to PRISM over most of the CONUS in all seasons, except that it shows wet biases over sporadic locations in NW and SW regions. The corresponding percent bias in Pmean24h

(Supplementary Fig. S8) indicates large Stage IV relative dry biases in the western CONUS (NGP, NW and SW) in DJF, possibly related to its inability to detect freezing and light precipitation events, as discussed in the previous subsection.

L284-290: The 24-hr mean wet-day precipitation (SDII) is shown in Fig. 13. As for the biases in Pmean24h (Fig. 12), Stage IV underestimates SDII almost everywhere, but more prominently over the eastern half of the CONUS in all seasons. ERA5 underestimates SDII over the eastern half of the CONUS (parts of NGP, MW, SGP, and NE) across the year. The dry SDII biases, together with the overestimated frequency and mean precipitation in winter and spring over NGP, MW, and SGP, suggest that ERA5 has too-little-and-too-frequent precipitation bias. WRF exhibits wet SDII biases over most of the CONUS in DJF, except in a few places over the SGP and SE. On the other hand, it shows strong dry biases over the SGP and SE during spring and over the SGP, MW and SE during summer.

L291-294: Fig. 14 shows biases in 24-hr annual maximum precipitation (Rx1day). As for the other metrics, Stage IV underestimates Rx1day over the eastern half of the CONUS. The dry bias is most pronounced ($\sim 20\%$) over the Great Plains and MW during summer and over the NGP and northeastern parts of SW ($> 50\%$) during winter (Supplementary Material Fig. S10). On the other hand, Rx1day values in Stage IV are very well represented over NW and SW in all seasons except winter.

L301-302: Finally, the PDF of 24-hr annual maximum precipitation (PDF24h)

16

is shown in Fig. 15. Stage IV represents well the PDF24h over NW and SW. However, it does show problems in capturing the PDF24h over the NGP throughout the year.

**comment #17. L159-160: sentence a little confusing I think "averaged precipitation peak are improved in the downscaled WRF simulations compared to ERA5" is a bit more clear.**

**Response:** The revised text in lines 189-190 reads as follows.

L192-193: In summary, both the timing and magnitude of the monthly averaged precipitation peak are improved in the downscaled WRF simulations compared to ERA5.

**comment #18. Figure 6 - what is the giant red dot in the Stage IV product? I think these figures would be improved if you masked out the red dot and you re-scaled precipitation ... all this blue makes things hard to see.**

**Response:** We have removed the spurious dot from the revised figures.

**comment #19. L174-177: Starting with "The maximum values ..." the comments about CMORPH seem really random and an aside given that you are not talking about CMORPH ...**

**Response:** We have retained the discussion on CMORPH to indicate that 0.25°0.25° grid spacing in CMORPH may not be sufficient for the reasonable representation of 3-hr precipitation mean along the NW US. Please refer to lines 207-210 of the revised manuscript.

**comment #20. Fig 7: Do you include zeros in this average? If**

17

so it might also be good to look at the precipitation rate when it rains.

**Response:** Zero precipitation values are included in the calculation of mean precipitation (Pmean3h and Pmean24h). Please see our response to your comment #3 above.

**comment #20. Fig 11/12 - should this be PRISM in the left and column?**

**Response:** Corrected in the revised manuscript.